# Mechanical sensing protein PIEZO1 regulates bone homeostasis via osteoblast-osteoclast crosstalk

Lijun Wang [1], Xiuling You[1], Sutada Lotinun[2], Lingli Zhang[1], Nan Wu [3] & Weiguo Zou [1,4]*

Wolff's law and the Utah Paradigm of skeletal physiology state that bone architecture adapts to mechanical loads. These models predict the existence of a mechanostat that links strain induced by mechanical forces to skeletal remodeling. However, how the mechanostat influences bone remodeling remains elusive. Here, we find that *Piezo1* deficiency in osteoblastic cells leads to loss of bone mass and spontaneous fractures with increased bone resorption. Furthermore, *Piezo1*-deficient mice are resistant to further bone loss and bone resorption induced by hind limb unloading, demonstrating that PIEZO1 can affect osteoblast-osteoclast crosstalk in response to mechanical forces. At the mechanistic level, in response to mechanical loads, PIEZO1 in osteoblastic cells controls the YAP-dependent expression of type II and IX collagens. In turn, these collagen isoforms regulate osteoclast differentiation. Taken together, our data identify PIEZO1 as the major skeletal mechanosensor that tunes bone homeostasis.

[1] State Key Laboratory of Cell Biology, CAS Center for Excellence in Molecular Cell Sciences, Shanghai Institute of Biochemistry and Cell Biology, Chinese Academy of Sciences, University of Chinese Academy of Sciences, Shanghai 200031, China. [2] Department of Physiology and Skeletal Disorders Research Unit, Faculty of Dentistry, Chulalongkorn University, Bangkok, Thailand. [3] Department of Orthopaedic Surgery, Peking Union Medical College Hospital, Peking Union Medical College and Chinese Academy of Medical Sciences, Beijing, China. [4] Institute of Microsurgery on Extremities, Shanghai Jiao Tong University Affiliated Sixth People's Hospital, Shanghai 200233, China. *email: zouwg94@sibcb.ac.cn

Bone remodeling involves the removal of old or damaged bone by osteoclasts (bone resorption) and subsequent replacement of new bone formed by osteoblasts (bone formation). One of the major functions of bone remodeling is to adapt bone material and structural properties to the mechanical demands that are placed on the skeleton, including mechanical loading and weight bearing[1]. Increased loading stimuli, e.g., through exercise and vigorous muscular activity, will enhance bone mass and bone strength. Alternatively, lack of mechanical loading in the setting of prolonged bed rest or exposure to long-term microgravity environment in space leads to a rapid reduction in bone mass and bone strength[2–4]. This mechanical force-bone interplay has been recognized as Julius Wolff's law since the 19th century[5,6]. Understanding how mechanical loading regulates the balance between bone formation and resorption would lead to the therapeutic strategies for disuse osteoporosis.

It has been reported that osteoblast lineage cells, including bone mesenchymal stem cells (BMSCs), osteoblast progenitor cells, osteoblasts, and osteocytes can respond to mechanical loading in bone[7]. TRPV4 mediates fluid shear stress induced calcium signaling and early osteogenic gene expression in BMSCs[8]. The opening of Connexin43 (Cx43) hemichannel depends on interaction with integrin in response to shear stress in osteocytes[9]. However, Cx43 deficient mice showed an enhanced anabolic response to mechanical load[10]. YAP and TAZ can sense a broad range of mechanical cues and transduce them into transcriptional response in a manner that is specific for each type of cell and mechanical stress, including ECM rigidity and topology, stretching and tensional forces[11]. The above molecules are important transducers of mechanical signals in skeletal cells and regulate the function of osteoblast linage cells. While mechanical loading promotes osteoblast differentiation, it also inhibits osteoclast formation, migration, and adhesion in vivo. However, how mechanical loading coordinates bone remodeling remains incompletely understood.

Mechanically activated cation channel activity has been recorded in many cells, though the responsible sensing molecules were mysterious until recently[12]. Expression profiling and RNA interference knockdown of candidate genes in a mouse neuroblastoma cell line identified PIEZO proteins (PIEZO1 and PIEZO2) as nonselective $Ca^{2+}$-permeable cation channels[13]. PIEZO proteins are emerging as important mediators of various aspects of mechanotransduction[14–16]. PIEZO1 is expressed by non-sensory tissues and could sense various mechanical stresses, including static pressure, shear stress, and membrane stretch[17–19]. PIEZO1 has been reported to mediate the function of hydrostatic pressure on cell fate determination of mesenchymal stem cells[19]. However, it is unknown whether PIEZO1 can regulate bone remodeling responding to mechanical load. In this study, we have generated Piezo1 conditional-knockout mice, and find that mice with Piezo1-deficiency in BMSCs and their progeny cells display increased bone resorption and multiple spontaneous fractures after weight bearing. Piezo1-deficient mice are resistant to further bone loss and osteoclast accumulation induced by mechanical unloading. Mechanistically, Piezo1 deficiency impairs COL2 and COL9 production through decreasing YAP nuclear localization, causing increased osteoclast activity. All of the above data support the notion that PIEZO1 can function as the mechanostat that directly senses mechanical loading to coordinate the osteoblast–osteoclast crosstalk in skeleton. Our study advances understanding and stimulates targeted therapeutic approaches for disuse osteoporosis.

## Results

**Osteoblastic *Piezo1* deficiency resulted in osteoporosis.** As PIEZO proteins are components of mechanically activated cation

channels, we hypothesize that PIEZO channels could function as the long sought mechanostat that directly senses mechanical loading in skeletal cells. If true, we predict that PIEZO channel deficiency would regulate bone mass and strength in vivo. QPCR confirmed that Piezo1 was highly expressed in bone and skeletal cells (Supplementary Fig. 1a–c), while Piezo2 was highly expressed in the dorsal root ganglia neurons instead of the whole bone and primary osteoblasts as reported[13,20] (Supplementary Fig. 1a–c). These results suggest that PIEZO1 could have a crucial role in osteoblasts. To further elucidate the function of PIEZO1 in the bone homeostasis, we generated a PIEZO1 conditional-knockout mouse model (Supplementary Fig. 1d, e) by crossing Piezo1[fl/fl] mice with Prx1[Cre] mice, which expressed Cre recombinase in osteoblast progenitors that form the limbs and parts of the skull, but not the spine or other organs in vivo[21]. QPCR confirmed a reduction of Piezo1 mRNA in bone cells from Prx1[Cre]; Piezo1[fl/fl] mice (Supplementary Fig. 1f). Immunofluorescence confirmed the deletion of PIEZO1 in osteoblast progenitors (Supplementary Fig. 1g). Yoda1 has been identified as a specific agonist for PIEZO1 but not PIEZO2[22]. Yoda1 stimulated calcium influx in wild-type, but not in Piezo1-deficient osteoblastic cells (Supplementary Fig. 1h). These data demonstrate functional deletion of PIEZO1 in osteoblast lineage cells in Prx1[Cre]; Piezo1[fl/fl] mice.

To determine in vivo effects of PIEZO1 within the skeletal system, we performed quantitative computed tomography (μ-QCT) analysis. Trabecular bone mass was significantly reduced in male Prx1[Cre]; Piezo1[fl/fl] mice compared to WT controls (Fig. 1a), as confirmed by decreased bone mineral density (BMD, Fig. 1b), trabecular bone volume (BV/TV, Fig. 1c), trabecular number (Tb. N, Fig. 1d) and increased trabecular spacing (Tb.Sp, Fig. 1f). Trabecular thickness (Tb.Th, Fig. 1e) was not changed significantly. In addition, cortical thickness was decreased in male Prx1[Cre]; Piezo1[fl/fl] mice (Ct.Th) (Fig. 1a, g). Female WT and Prx1[Cre]; Piezo1[fl/fl] mice were also analyzed by μ-QCT. The difference between WT and Piezo1-deficient female mice was comparable to male (Supplementary Fig. 2a–f), indicating that the function of Piezo1 is independent of gender. Furthermore, the long bones of Prx1[Cre]; Piezo1[fl/fl] mice were smaller than WT mice (Supplementary Fig. 2g). We also analyzed the bone surfaces of WT and Prx1[Cre]; Piezo1[fl/fl] mice by μ-QCT. Both cortical and trabecular bone surfaces of Prx1[Cre]; Piezo1[fl/fl] mice were significantly decreased, compared with WT mice (Supplementary Fig. 2h, i). Notably, we observed multiple bone fractures in the Prx1[Cre]; Piezo1[fl/fl] mice in weight-bearing appendicular bones (Fig. 1h). The fractures first occurred between P0 and P3, with no significant differences observed between wild-type and Piezo1-deficient mice at embryonic day 17.5 (Supplementary Fig. 3a) and postnatal day 0 (Supplementary Fig. 3b). However, multiple bone fractures occurred in 3-day-old Prx1[Cre]; Piezo1[fl/fl] mice (Fig. 1i, Supplementary Fig. 3c). The femurs of 3-day-old Piezo1-deficient mice exhibited increased cortical porosity compared to WT control mice by μ-CT analysis (Fig. 1j, bottom panel). However, neonatal P0 mice showed only subtle differences between Prx1[Cre]; Piezo1[fl/fl] mice and WT controls (Fig. 1j, top panel). Embryonic mice are surrounded by amniotic fluid, and therefore bones are not weight bearing. Therefore, we hypothesize that the absence of mechanical loading makes the effects of Piezo1 deficiency unseen at this stage. Overall the kinetics of phenotypic onset is consistent with PIEZO1 functioning downstream of postnatal mechanical loading. In addition, calvarial bones from WT and Prx1[Cre]; Piezo1[fl/fl] mice were indistinguishable (Supplementary Fig. 4a, b), compared to more dramatic differences in the distal femurs, perhaps due to the fact that the skull is relatively under-loaded compared to long bones. Collectively, these data support that Piezo1 deficiency impairs the response of osteoblastic cells to

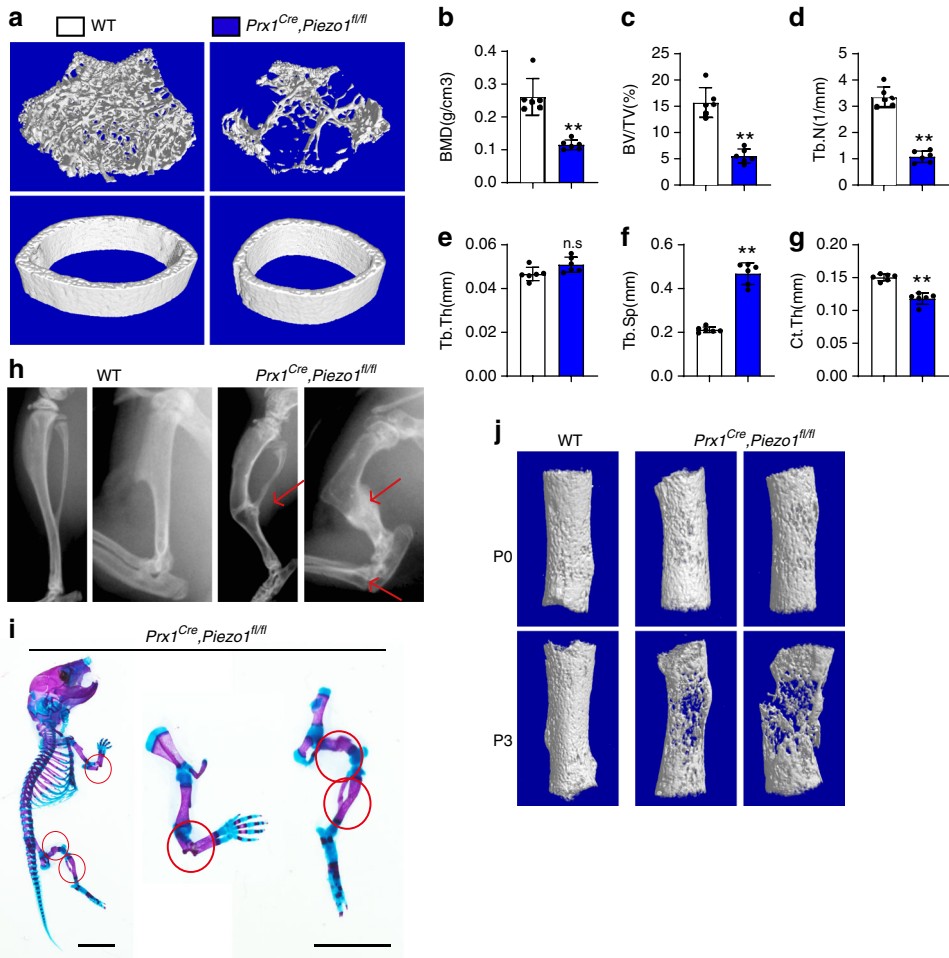

**Fig. 1 Loss of *Piezo1* in skeletal cells resulted in severe osteoporosis. a** 3D μ-CT images of trabecular bones of distal femurs isolated from 6-week-old male WT and *Prx1Cre, Piezo1fl/fl* mice. **b–g** μ-CT analysis of distal femurs from (**a**) for bone mineral density (BMD) (**b**), bone volume per tissue volume (BV/TV) (**c**), trabecular number (Tb.N) (**d**), trabecular thickness (Tb.Th) (**e**), trabecular spacing (Tb.Sp) (**f**) and cortical thickness (Ct.Th) of middle shaft of femurs (**g**). *$P < 0.05$; **$P < 0.01$. Two-tailed Student's $t$ test. Data are mean ± SD, $n = 6$. **h** X-ray of long bones from 6-week-old male WT and *Prx1Cre, Piezo1fl/fl* mice. Representative images for 3 independent samples. **i** Whole mount skeleton staining of 3-day-old WT and *Prx1Cre, Piezo1fl/fl* mice by Alcian blue and Alizarin red S. Scale bar = 5 mm. **j** 3D μ-CT images of femurs isolated from 0-day-old and 3-day-old WT and *Prx1Cre, Piezo1fl/fl* mice. Source data are provided in the Source Data File.

mechanical loading, leading to decreased bone mass and giving rise to bone fractures soon after birth.

**Osteoblastic *Piezo1* deficiency promoted bone resorption.** We next sought to understand the relative contributions of osteoblast and osteoclast activity to the bone loss seen in *Prx1Cre; Piezo1fl/fl* mice. Osteoblasts originate from SSCs (skeletal stem cells), Pre-BCSPs (pre-bone, cartilage and stromal progenitors), and BCSPs (bone, cartilage and stromal progenitors), whose expansion and self-renewal must be tightly controlled[23]. Firstly, we analyzed these three cell types in *Prx1Cre; Piezo1fl/fl* and control mice. However, these stem cells and progenitors were not significantly affected by PIEZO1 deficiency (Fig. 2a, b). We further speculated if PIEZO1 would impair osteoblast differentiation. To test this hypothesis, we performed histomorphometric analysis of WT and *Prx1Cre; Piezo1fl/fl* mice to evaluate static and dynamic parameters of bone formation and resorption. Consistent with the μ-QCT data, histomorphometric analysis showed that *Prx1Cre; Piezo1fl/fl* mice had a significant decrease in both BV/TV and Tb.N (Fig. 2c–e). Interestingly, the number of osteoblasts per bone perimeter (N.Ob/B.Pm) (Fig. 2f)

and osteoblast surface per bone surface (Ob.S/BS) (Fig. 2g) were not significantly changed in *Prx1Cre; Piezo1fl/fl* mice compared to WT controls. Consistently, the bone mineral apposition rate and bone formation rate were not significantly changed in *Prx1Cre; Piezo1fl/fl* mice either (Fig. 2h–k). Furthermore, the bone formation marker PINP was not significantly changed in the serum of *Prx1Cre; Piezo1fl/fl* mice (Fig. 2l). However, there was significantly increased bone resorption activity in *Prx1Cre; Piezo1fl/fl* mice compared to WT controls, demonstrated by increased eroded surface normalized to bone surface (ES/BS) (Fig. 2m), osteoclast number (N.Oc/Bpm) (Fig. 2n), osteoclast surface per bone surface (Oc.S/BS) (Fig. 2o) and serum CTX-I level (Fig. 2p) in *Prx1Cre; Piezo1fl/fl* mice. Tartrate-resistant acid phosphatase (TRAP) staining confirmed the increase of osteoclasts along the surface of trabecular bone in *Prx1Cre; Piezo1fl/fl* mice (Fig. 2q). Also, *Piezo1* deficiency did not impact osteoblast differentiation as determined by ALP activity and Alizarin red S staining in vitro (Supplementary Fig. 5a, b). Consistently, the expression of a series of osteoblastic marker genes showed no significant difference between control and *Piezo1*-deficient cells (Supplementary Fig. 5c).

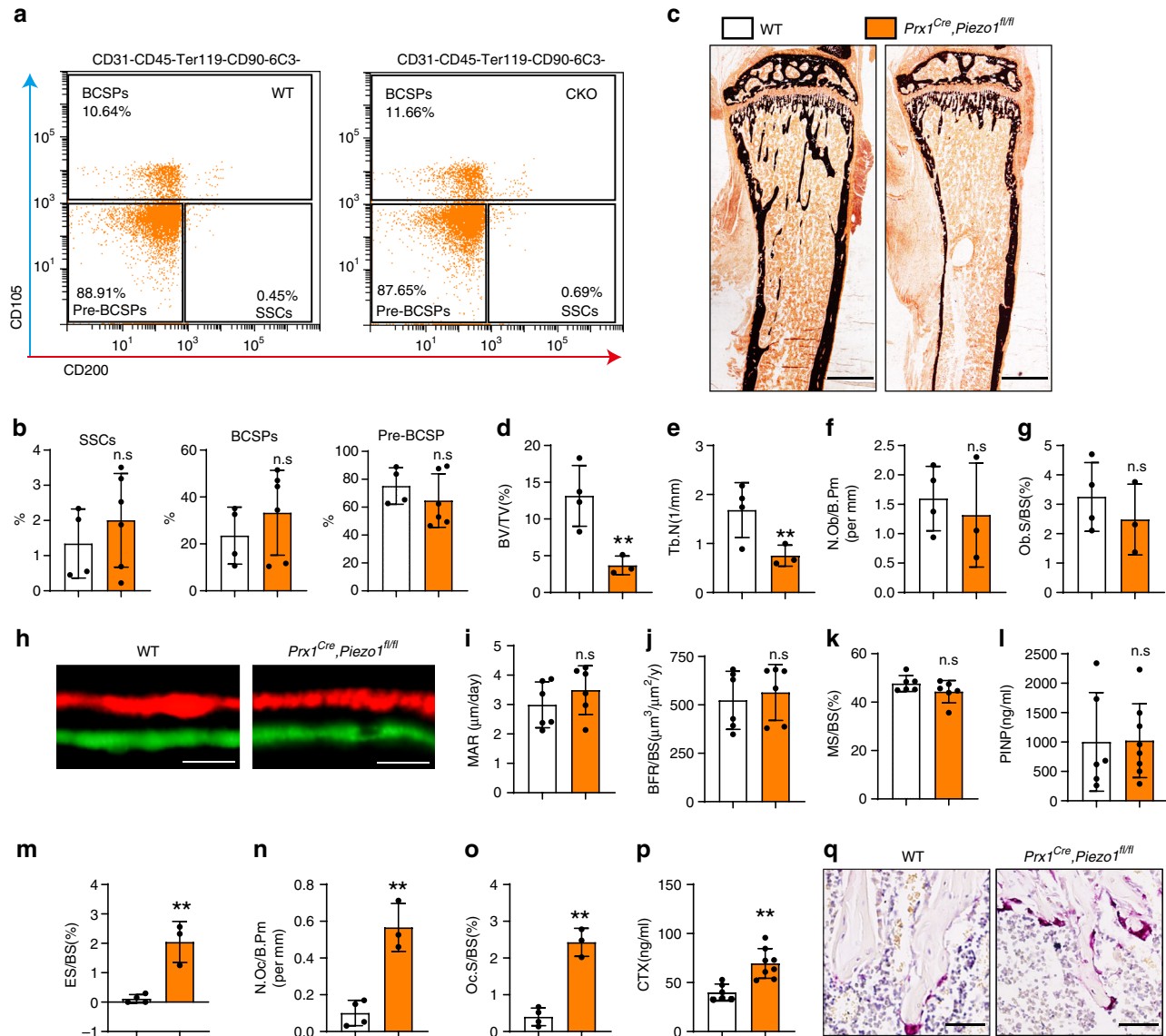

**Fig. 2 Loss of *Piezo1* in osteoblastic cells resulted in increased bone resorption. a**, **b** FACS analysis of mSSCs, mBSCPs and mPre-BSCPs of bone marrow cells of 6-week-old male WT and *Prx1^Cre^, Piezo1^fl/fl^* mice. Data are mean ± SD, WT, *n* = 4, CKO, *n* = 6. **c** Representative images of Von kossa staining of 6-week-old male WT and *Prx1^Cre^, Piezo1^fl/fl^* mice. Scale bar = 500 μm **d**–**g** Histomorphometric analysis of proximal tibias from 6-week-old male WT and *Prx1^Cre^, Piezo1^fl/fl^* mice for bone volume per tissue volume (BV/TV) (**d**), trabecular number (Tb.N) (**e**), number of osteoblasts per bone perimeter (N.Ob/B. Pm) (**f**), osteoblast surface per bone surface (Ob.S/BS) (**g**). Data are mean ± SD, WT, *n* = 4, CKO, n = 3. **h** Representative images of dual Calcein-Alizarin red S labeling of proximal tibias from 6-week-old male WT and *Prx1^Cre^, Piezo1^fl/fl^* mice. Scale bar = 20 μm, n = 6. **i**–**k** Quantification of mineral apposition rate (MAR) (**i**), bone formation rate normalized to bone surface (BFR/BS) (**j**) and mineralizing surface normalized to bone surface (MS/BS) (**k**) of endo-cortical and trabecular bones of proximal tibias from 6-week-old male WT and *Prx1^Cre^, Piezo1^fl/fl^* mice. Data are mean ± SD, *n* = 6. **l** Serum PINP levels in 6-week-old male WT and *Prx1^Cre^, Piezo1^fl/fl^* mice. Data are mean ± SD, WT, *n* = 6, CKO, *n* = 8. **m**–**o** Histomorphometric analysis of the images from (**c**) for eroded surface per bone surface (ES/BS) (**m**), number of osteoclasts per bone perimeter (N.Oc/B.Pm) (**n**) and osteoclast surface per bone surface (Oc.S/BS) (**o**). Data are mean ± SD, WT, *n* = 4, CKO, *n* = 3. **p** Serum CTX-I levels in 6-week-old male WT and *Prx1^Cre^, Piezo1^fl/fl^* mice. Data are mean ± SD, WT, *n* = 6, CKO, *n* = 8. **q** Tartrate-resistant acid phosphatase (TRAP) staining of femurs isolated from 6-week-old male WT and *Prx1^Cre^, Piezo1^fl/fl^* mice. Scale bar = 50 μm. *$P < 0.05$; **$P < 0.01$. Two-tailed Student's *t* test. Source data are provided in the Source Data File.

Osteocytes, bone cells embedded in the bone matrix, are thought to be critical mechanosensory cells in bone. *Dmp1^Cre^; Piezo1^fl/fl^* mice, which deleted PIEZO1 in osteocytes, exhibited decreased bone mass in both trabecular and cortical bones, compared with control mice (Supplementary Fig. 6a–g). In addition, osteoclast number was also increased in *Dmp1^Cre^; Piezo1^fl/fl^* mice (Supplementary Fig. 6h, i). However, different from *Prx1^Cre^; Piezo1^fl/fl^* mice, no spontaneous fractures occurred in *Dmp1^Cre^; Piezo1^fl/fl^* mice (Supplementary Fig. 6j), indicating

that not only osteocytes but also earlier osteoblastic cells are essential for mechanical sensing in bone.

**Osteoclastic *Piezo1* deficiency did not affect bone mass.** In order to test if PIEZO1 affects the osteoclast function, we crossed *Piezo1^fl/fl^* mice with *Ctsk^Cre^* mice to delete *Piezo1* in osteoclasts. *Ctsk^Cre^; Piezo1^fl/fl^* mice exhibited normal bone mass (Fig. 3a–c) and unaffected bone resorption (Fig. 3d, e), compared with

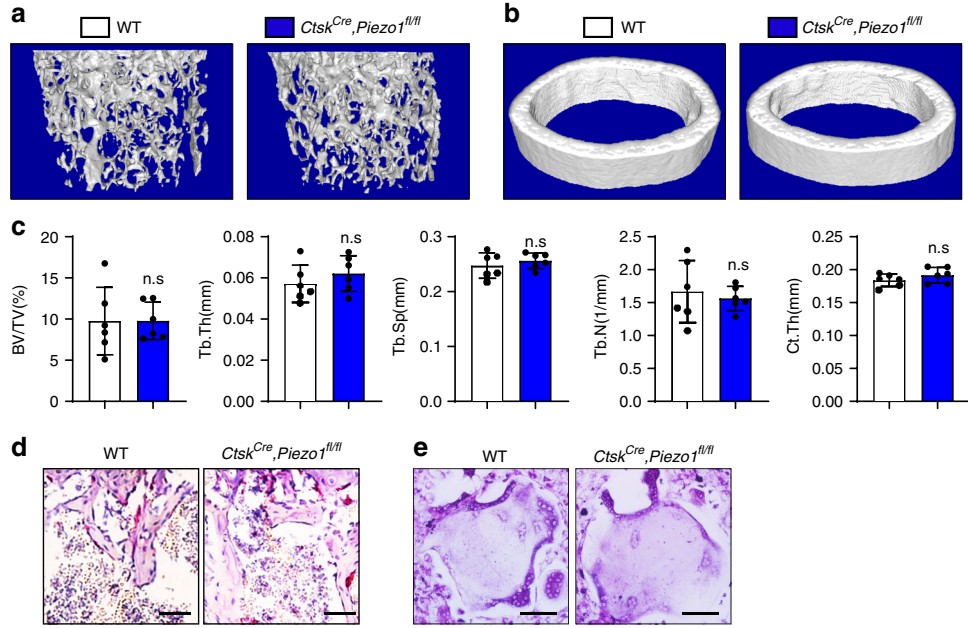

**Fig. 3 *Piezo1* deficiency in osteoclasts did not affect the bone formation. a**, **b** 3D μ-CT images of trabecular bones (**a**) and cortical bones (**b**) of distal femurs isolated from 6-week-old female WT and *Ctsk^Cre, Piezo1^fl/fl* mice. **c** μ-CT analysis of distal femurs from 6-weeks-old female WT and *Ctsk^Cre, Piezo1^fl/fl* mice for bone volume per tissue volume (BV/TV), trabecular thickness (Tb.Th), trabecular number (Tb.N), trabecular spacing (Tb.Sp) and cortical thickness (Ct.Th) of middle shaft of femurs. *$P < 0.05$; **$P < 0.01$. Two-tailed Student's $t$ test. Data are mean ± SD, $n = 6$. **d** TRAP staining of femurs isolated from 6-week-old WT and *Ctsk^Cre, Piezo1^fl/fl* mice. **e** TRAP staining of osteoclasts differentiated from bone marrow monocytes isolated from WT and *Ctsk^Cre, Piezo1^fl/fl* mice. Scale bar = 100 μm. Source data are provided in the Source Data File.

control mice. These data indicate that PIEZO1 acts primarily in osteoblastic cells to orchestrate bone resorption by osteoclasts in a non-cell autonomous manner.

**PIEZO1 sensed the weight-bearing induced mechanical stress**. Although bone resorption was strongly increased in the long bones of *Prx1^Cre; Piezo1^fl/fl* mice, TRAP positive osteoclasts were not significantly changed in skull whole mounts of *Prx1^Cre; Piezo1^fl/fl* mice (Supplementary Fig. 7a). Moreover, TRAP positive osteoclasts in *Prx1^Cre; Piezo1^fl/fl* mice were only modestly increased at P0, but were more substantially expanded on postnatal day 3 (Supplementary Fig. 7b–e), consistent with the kinetics of postnatal weight bearing.

To further determine if PIEZO1 mediates skeletal mechanical loading, tail suspension experiments were performed to remove body weight-induced mechanical loading from hind limbs, mimicking the bone loss due to microgravity or disuse[24,25]. As expected, tail suspension induced bone loss in the distal femurs of WT mice (Fig. 4a, left panel). Strikingly, *Prx1^Cre; Piezo1^fl/fl* mice did not demonstrate tail suspension induced bone loss (Fig. 4a–f). Consistently, tail suspension treatment increased TRAP positive cells in WT but not in *Prx1^Cre; Piezo1^fl/fl* mice (Fig. 4g, h). These data support that PIEZO1 in osteoblastic cells is required to sense mechanical loading and accordingly regulate bone mass by modulating the bone resorption activity of osteoclasts.

**Collagens mediated inhibition of bone resorption by PIEZO1**. We next investigated the molecular mechanisms through which *Piezo1* deficiency in osteoblast lineage cells controls bone homeostasis. BMSCs-derived osteoblasts from *Prx1^Cre, Piezo1^fl/fl* mice showed an enhanced ability to support osteoclastogenesis, as indicated by the increased number of giant multinucleated cells (Fig. 5a) and TRAP activity of culture supernatants (Fig. 5b) in an osteoblast–osteoclast co-culture system. To determine whether

osteoclasts are more active or whether there are more osteoclasts with similar activity, we performed pit resorption assay in osteoblast–osteoclast co-culture system. Osteoclasts in the *Piezo1*-deficient osteoblastic cells co-culture system are increased in number and more active in resorption, compared with control osteoblastic cells (Fig. 5c, d). These data demonstrate that increased bone resorption in *Piezo1*-deficient mice results from both increased osteoclast number and activity.

We next screened the key factors secreted by osteoblasts to tune the activity of osteoclasts, including *RankL, Opg, Sost, Pthlh, Mcsf* and *Sema3a*[26]. However, none of these factors showed a significant difference in the cortical bones of WT and *Prx1^Cre; Piezo1^fl/fl* mice (Fig. 5e). Therefore, we performed RNA-Seq using RNA from tibial and femoral cortical bones of WT and *Prx1^Cre; Piezo1^fl/fl* mice. As shown in Supplementary Figure 8a, the global transcriptome was changed between wild-type and *Piezo1*-deficient mice. Among a total of 19,201 genes expressed, 560 genes were up-regulated, and 297 genes were down-regulated (fold change > 1.5, $P$. value < 0.05) in *Piezo1*-depleted cortical bones. We performed GO analysis and found that the genes of proteinaceous extracellular matrix were significantly down regulated (Supplementary Fig. 8b). Among these matrix proteins, COL2α1, COL9α1/2 and COL10α1 were decreased substantially in *Prx1^Cre; Piezo1^fl/fl* mice (Supplemental Fig. 8c). However, there was no significant difference in type 1 collagen. QPCR further validated the changes of these genes in the cortical bones of WT and *Prx1^Cre; Piezo1^fl/fl* mice (Fig. 5f). Immunofluorescence analysis confirmed the changes of protein levels of different collagens including COL2α1 (Fig. 5g) and COL9α2 (Fig. 5h). These data indicate that *Piezo1* deficiency changes the integrity of the bone matrix, especially of multiple collagens.

To directly examine if PIEZO1 acts as downstream of mechanical stimulation to regulate collagen expression, BMSCs-derived osteoblasts from *Prx1^Cre, Piezo1^fl/fl* and WT mice were cultured in the Flexcell Compression System[27]. The expression

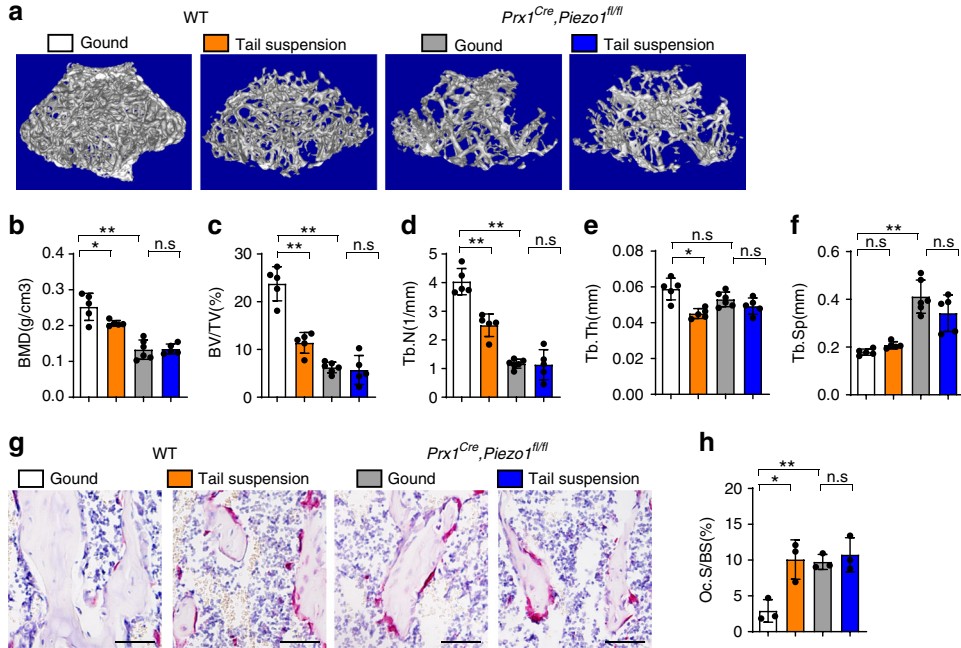

**Fig. 4 *Piezo1* deficiency resisted to bone loss induced by unloading. a** 3D μ-CT images of trabecular bones of distal femurs isolated from male WT and *Prx1^Cre*, *Piezo1^fl/fl* mice, subjected with tail-suspended for one week trial or ground control, respectively, sacrificed at 7-week-old. **b–f** μ-CT analysis of distal femurs for bone mineral density (BMD) (**b**), bone volume per tissue volume (BV/TV) (**c**), trabecular number (Tb.N) (**d**), trabecular thickness (Tb.Th) (**e**) and trabecular spacing (Tb.Sp) (**f**). Data are mean ± SD, n ≥ 5. **g** TRAP staining of the femurs isolated from male WT and *Prx1^Cre*, *Piezo1^fl/fl* mice, subjected with tail-suspended trial or ground control, respectively. Scale bar = 50 μm. **h** Quantification of the TRAP staining by osteoclast surface per bone surface (Oc.S/BS). Data are mean ± SD, n = 3. *P < 0.05; **P < 0.01. Ordinary one-way ANOVA. Source data are provided in the Source Data File.

levels of *Col2α1* (Fig. 5i) and *Col9α2* (Fig. 5j) were strongly increased by the compression system in WT cells, compared with *Piezo1*-deficient cells. Matrix proteins including COL2 and COL9 can inhibit osteoclast activity[28,29], indicating that PIEZO1 could control bone homeostasis through the regulation of bone matrix proteins including *Col2α1* and *Col9α2* expression. Next, we co-cultured bone marrow monocytes with WT and *Piezo1*-deficient osteoblastic cells overexpressing GFP control, COL2α1 or COL9α2. *Piezo1*-deficient osteoblastic cells increased the osteoclast differentiation compared with WT cells, while both COL2α1 and COL9α2 repressed this increased differentiation of osteoclasts, evidenced by TRAP staining and supernatants TRAP activity (Fig. 5k–l). These data indicate that type II and IX collagens serve as key effectors in PIEZO1-mediated mechanotransduction in bone.

**PIEZO1 promoted *Col2α1* and *Col9α2* expression through YAP.** We then investigated how PIEZO1-linked mechanical loading to the expression of different collagens. It has been known that PIEZO1 regulates the nuclear localization of YAP[30]. Decreased YAP nuclear localization in *Piezo1*-deficient cells was confirmed by immunofluorescence (Fig. 6a, b). We next sought to determine the effects of YAP deficiency on the expression of different collagens. Knockdown of YAP by two different shRNAs lead to the decrease of *Col2α1* and *Col9α2* expression (Fig. 6c, d). Moreover, XMU-MP-1, a MST1/2 kinase inhibitor, which activates the downstream effector YAP[31], increased the expression of *Col2α1* and *Col9α2* (Fig. 6e). In addition, the promoter of *Col2α1* and *Col9α2* could be activated by YAP (Fig. 6f, g). To further demonstrate the functional relationship between PIEZO1 and YAP, we treated the co-culture system with XMU-MP-1 to activate YAP. *Piezo1*-deficient osteoblastic cells could promote the osteoclast differentiation, while XMU-MP-1 could inhibit the increased osteoclast differentiation caused by *Piezo1* deficiency

(Fig. 6h, i). To exclude the function of XMU-MP-1 on osteoclasts, we treated bone marrow monocytes with XMU-MP-1 during osteoclast differentiation. XMU-MP-1 did not significantly affected the osteoclast formation (Supplementary Fig. 9a, b). Overall, these data demonstrate a PIEZO1-YAP pathway controlling the expression of type II and IX collagens in osteoblast lineage cells thereby affecting osteoclastogenesis.

**Inducible osteoblastic *Piezo1* deletion decreased bone mass.** To mimicking sudden loss of gravity or weight-bearing in a space flight or long-term bedridden, we utilized tamoxifen-inducible osteoblast lineage cells expressing Cre, *Col1^cre ert2* to induce *Piezo1* deletion at 8-week-old for 2 weeks (Fig. 7a). To determine the in vivo effects of *Piezo1* in this system, we performed μ-QCT analysis and found that trabecular bone mass was significantly reduced in *Col1^Cre ert2*; *Piezo1^fl/fl* mice compared to WT controls (Fig. 7b), as confirmed by decreased trabecular bone volume (BV/TV, Fig. 7c), trabecular number (Tb.N, Fig. 7d), trabecular thickness (Tb.Th, Fig. 7e) and increased trabecular spacing (Tb.Sp, Fig. 7f). Also, cortical thickness (Ct.Th.) was decreased in *Col1^Cre ert2*; *Piezo1^fl/fl* mice (Fig. 7b, g). Consistent with our previous findings in *Prx1^Cre*, *Piezo1^fl/fl* mice, TRAP positive osteoclasts were increased in *Col1^Cre ert2*; *Piezo1^fl/fl* mice (Fig. 7h, i). Also, collagen expression was compromised in this sudden *Piezo1*-loss model (Fig. 7j). In conclusion, our data demonstrate that sudden loss of the mechanosensor *Piezo1* in osteoblast lineage cells can modulate bone remodeling and mechanotransduction by PIEZO1 is essential for maintaining the bone homeostasis.

**Discussion**

Our current study found that PIEZO1 acts in osteoblast lineage cells as a regulator of bone remodeling via sensing mechanical loading. First, *Piezo1*-deficient osteoblastic cells promote bone

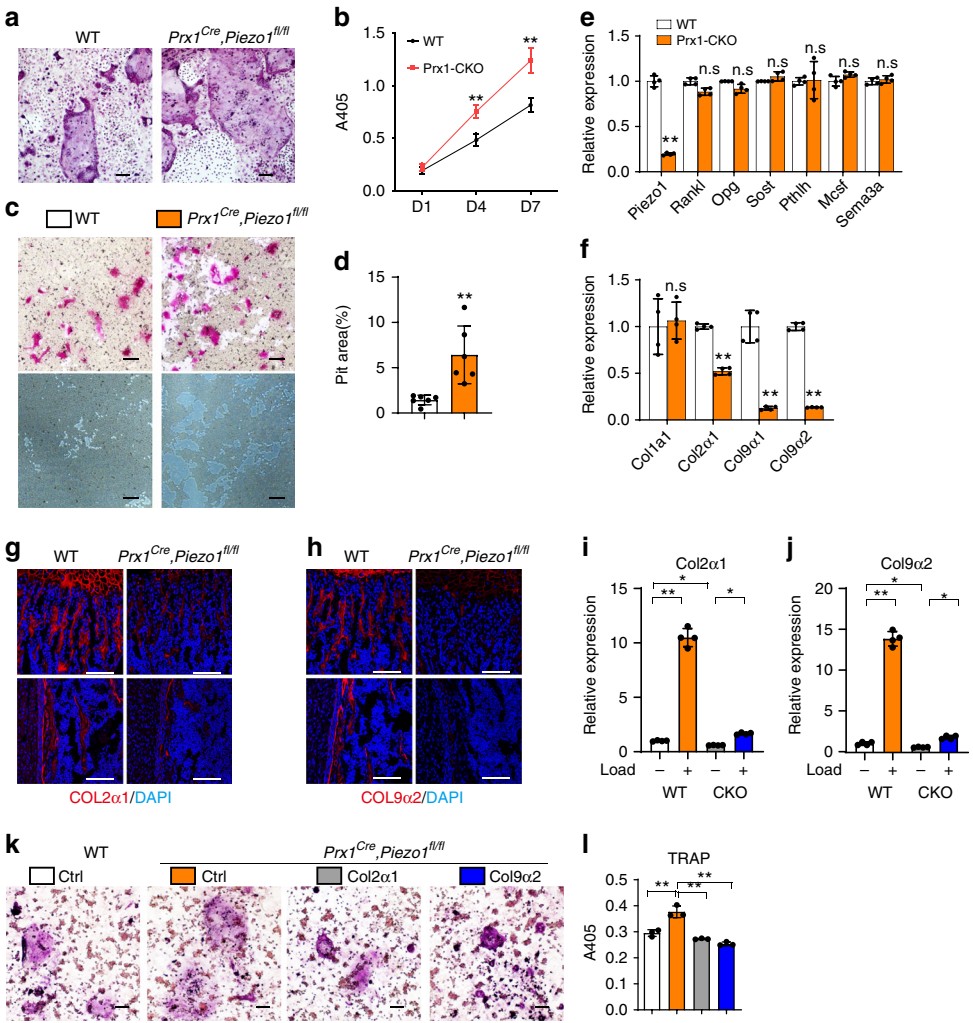

**Fig. 5 Col2α1 and Col9α2 mediated the inhibition of bone resorption by PIEZO1. a–d** Osteoclastogenesis by OB-OC co-culture in vitro using BMSCs-derived osteoblasts from WT and *Prx1^Cre^, Piezo1^fl/fl^* mice. Scale bar = 100 μm. **a** TRAP staining of the osteoclasts in the co-culture system on the normal surface. **b** Co-culture supernatants were measured for TRAP activity via colorimetric readout (A405) on the normal surface. *P < 0.05; **P < 0.01. Two-tailed Student's *t* test. Data are mean ± SD, n = 5. **c** TRAP staining of the osteoclasts (top panel) and resorption area (bottom panel) in the co-culture system on the bone biomimetic synthetic surface. **d** Quantification of the resorption area. *P < 0.05; **P < 0.01. Two-tailed Student's *t* test. Data are mean ± SD, n = 6. **e, f** Indicated gene expression analysis of the cortical bones of WT and *Prx1^Cre^, Piezo1^fl/fl^* mice. *P < 0.05; **P < 0.01. Two-tailed Student's *t* test. Data are mean ± SD, n = 4. (**g, h**) Immunofluorescence assay of COL2α1 (**g**) and COL9α2 (**h**) of the distal femurs of 3-day-old WT and *Prx1^Cre^, Piezo1^fl/fl^* mice. Scale bar = 100 μm. **i, j** QPCR analysis of *Col2α1* (**i**) and *Col9α2* (**j**) in the BMSCs-derived osteoblasts from WT and *Prx1^Cre^, Piezo1^fl/fl^* mice endured with 0.5 Hz, 1% intensity compression for 4 h by FlexCell compression system. *P < 0.05; **P < 0.01. Ordinary one-way ANOVA. Data are mean ± SD, n = 4. **k–l** Osteoclastogenesis by OB-OC co-culture in vitro using BMSCs-derived osteoblasts from WT and *Prx1^Cre^, Piezo1^fl/fl^* mice infected with *Ctrl*, *Col2α1* and *Col9α2* lenti-virus, respectively. **k** TRAP staining of the osteoclasts in the co-culture system. Scale bar = 100 μm. **l** Co-culture supernatants were measured for TRAP activity via colorimetric readout (A405). *P < 0.05; **P < 0.01. Ordinary one-way ANOVA. Data are mean ± SD, n = 3. Source data are provided in the Source Data File.

resorption by osteoclasts, accounting for the decreased bone mass of *Piezo1*-deficient mice. This phenotype is similar to the one of mechanical unloading with the increased bone resorption[32]. Secondly, our mechanism study demonstrates that PIEZO1 regulates YAP signaling, which in turn regulates the expression of different bone matrix proteins including several collagens. The expression of these collagens occurs through a PIEZO1-dependent response to mechanical stimulation. Taken together, these data identify the molecular basis for observations made in Wolff's Law and the Utah Paradigm linking the mechanical forces acting on bone to bone homeostasis[5,6]. Interestingly, an atlas of genetic influences on osteoporosis in humans identified several SNPs of *PIEZO1* associated with osteoporosis and fractures[33], indicating that PIEZO1 could also regulate bone mass and bone strength in humans in vivo.

Our study found that PIEZO1 regulates YAP-dependent expression of collagens that in turn regulate bone resorption. Type II collagen is encoded by *Col2α1* gene, and synthesized as a pro-collagen form with extension pro-peptides (NH2-pro-peptide and COOH-pro-peptide) that are removed by specific proteinases before the mature molecules are incorporated into fibrils in matrix[34]. The NH2-pro-peptide of type II collagen (PIIBNP) removed by ADAMTS-3[35] can bind to osteoclasts via an RGD-mediated manner to inhibit bone resorption[28]. Type IX collagen-deficient mice exhibit osteoporosis with normal bone formation and increased bone resorption[29]. Loss of collagen IX results in abnormal nano-topography of bone, and this abnormal surface can increase the size and resorption area of osteoclasts[29]. In addition, type IX collagens also contain RGD sequences predicted to bind integrin receptors. However, the precise receptors on

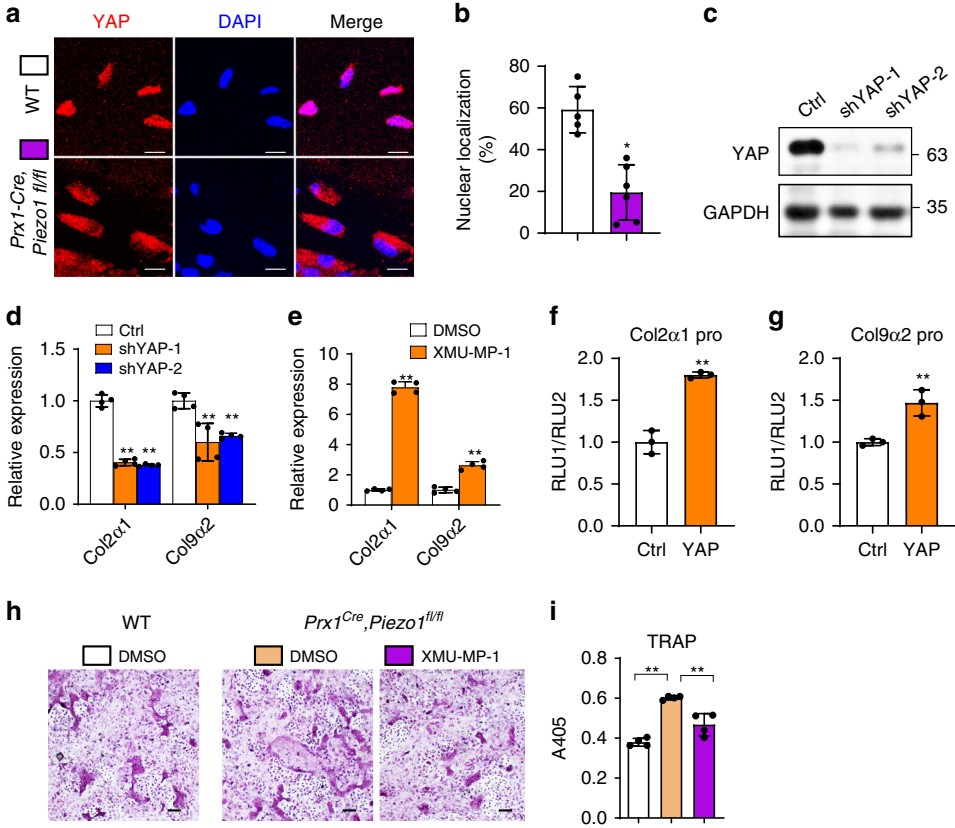

**Fig. 6 PIEZO1 promoted collagen expression through YAP nuclear localization. a** Immunofluorescence assay of YAP in the cortical bones of WT and *Prx1^Cre, Piezo1^{fl/fl}* mice. Scale bar = 10 μm. **b** Quantification of YAP nuclear localization percentage in WT and *Prx1^Cre, Piezo1^{fl/fl}* mice. *$P < 0.05$. Two-tailed Student's *t* test. Data are mean ± SD, $n ≥ 5$. **c, d** YAP expression by western blot (**c**) and indicated gene expression analysis (**d**) in the C3H10 cells infected with shControl and shYAP lenti-virus. **e** Indicated gene expression analysis in the C3H10 cells treated with DMSO or XMU-MP-1 (10 μm) for 24 h. **f, g** Luciferase assays showed that YAP could activate *Col2α1* (**f**) and *Col9α2* (**g**) promoters in the C3H10 cells. **h, i** Osteoclastogenesis by OB-OC co-culture in vitro using BMSCs-derived osteoblasts from WT and *Prx1^Cre, Piezo1^{fl/fl}* mice, the co-culture system was treated with DMSO or XMU-MP-1 (0.2 μm), respectively. **h** TRAP staining of the osteoclasts in the co-culture system. Scale bar = 100 μm. **i** Co-culture supernatants were measured for TRAP activity via colorimetric readout (A405). *$P < 0.05$; **$P < 0.01$. Ordinary one-way ANOVA. Data are mean ± SD, $n = 4$. Source data are provided in the Source Data File.

osteoclasts responding to these type II and IX collagen 'ligands' remain to be defined. Integrins are heterodimeric adhesion receptors that mediate cell–matrix and cell–cell interactions[36]. Integrin αvβ3 and α2β1 are two integrin receptors highly expressed on osteoclasts and can respond to RGD-containing ligands in the bone matrix to regulate osteoclast differentiation and function[37–40]. We proposed that integrin could mediate the repression of collagen on osteoclastogenesis. To test this hypothesis, we enriched the COL2α1 and COL9α2 proteins from the supernatants of *Col2α1* and *Col9α2* overexpressing 293T cells and treated the bone marrow monocytes during osteoclast differentiation. Both COL2α1 and COL9α2 proteins could inhibit osteoclastogensis (Supplementary Fig. 10a–c), consistent with co-culture experiments (Fig. 5k-l). However, COL2α1 and COL9α2 proteins could not repress osteoclast formation when the cells were treated with integrin inhibitors SB273005 or RGD peptides (Supplementary Fig. 10d–g). Our experiments suggest that integrins may serve as promising candidates for bridging matrix and osteoclast regulation.

Recent discoveries are converging on a model in which multiple types of mechanical inputs in a variety of cellular settings rely on the regulation of two transcriptional regulators, YAP and TAZ[11]. YAP/TAZ activity is regulated by the conformation and tension of the F-actin cytoskeleton, which in turn depends primarily on the substrate to which cells adhere. In addition, YAP

nucleo–cytoplasmic localization is influenced by PIEZO1 in human neural stem/progenitor cells[30]. Combinatorial YAP/TAZ deletion from skeletal lineage cells caused an osteogenesis imperfecta-like phenotype through regulation of osteoblast activity, matrix quality, and osteoclastic remodeling[41]. In our study, decreased YAP nuclear localization was observed in *Piezo1*-deficient mice, resulting in impaired collagen II and IX expression which leads to increased bone resorption. YAP/TAZ and PIEZO1 serve as downstream effectors of Wnt5a-mediated actomyosin polarity and cytosolic calcium transients that orient and drive three-dimensional cell intercalations that shape the murine mandibular arch[42]. It remains to be investigated whether Wnt5a will regulate the PIEZO1-YAP-collagen pathway in mechanotransduction mediated bone remodeling.

The skeletal system provides the mechanical framework for the whole body. The reduction of bone mass due to aging or hormone related osteoporosis could be treated by different strategies including anabolic treatments to increase bone formation and catabolic treatments to decrease bone resorption[43]. Indeed, astronauts typically lose more bone mass during one month than postmenopausal women on Earth lose in 1 year, at approximately 1–2% per month, with an even greater decrease in bone strength[44]. However, bone loss due to microgravity or the absence of mechanical forces cannot be effectively suppressed[45], indicating that the molecular mechanisms underlying bone loss from

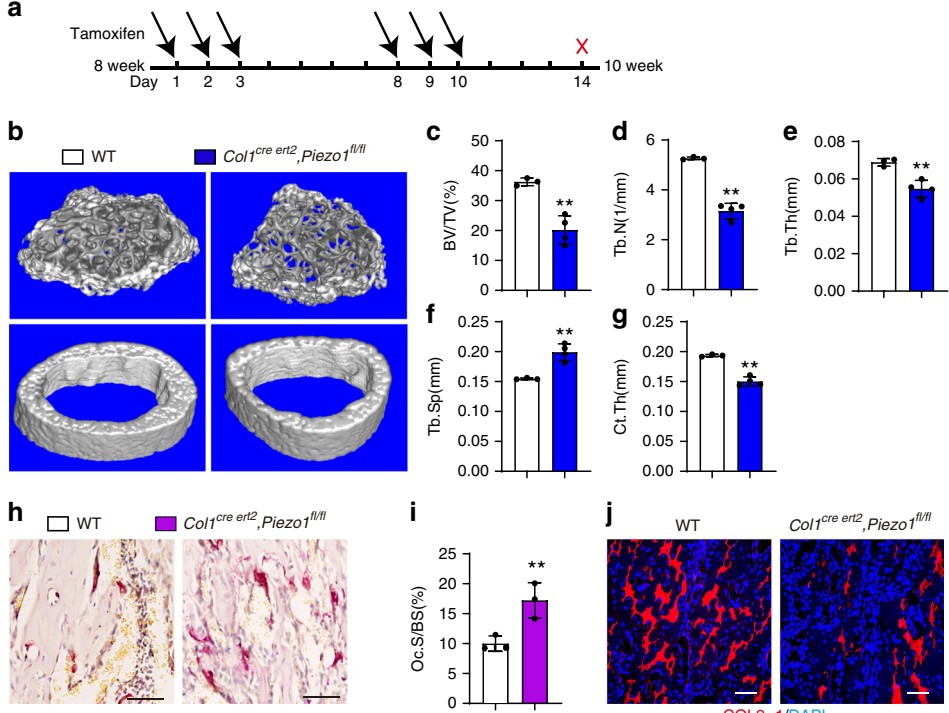

**Fig. 7 Sudden loss of *Piezo1* in osteoblast lineage cells decreased the bone mass. a** Scheme of the experiment: male WT and *Col1^Cre ert2^, Piezo1^fl/fl^* mice were injected with tamoxifen (0.1 g/kg) at 8-week-old and sacrificed for analysis at 10-week-old. **b** 3D μ-CT images of trabecular bones of distal femurs. **c–g** μ-CT analysis of distal femurs for bone volume per tissue volume (BV/TV) (**c**), trabecular number (Tb.N) (**d**), trabecular thickness (Tb.Th) (**e**), trabecular spacing (Tb.Sp) (**f**) and cortical thickness (Ct.Th) of middle shaft of femurs (**g**). Data are mean ± SD, WT, n = 3, CKO, n = 4. **h** TRAP staining of the femurs isolated from 10-week-old male WT and *Col1^Cre ert2^, Piezo1^fl/fl^* mice. Scale bar = 50 μm. **i** Quantification of the TRAP staining by osteoclast surface per bone surface (Oc.S/BS). Data are mean ± SD, n = 3. **j** Immunofluorescence assay of COL2α1 of 10-week-old male WT and *Col1^Cre ert2^, Piezo1^fl/fl^* mice. Scale bar = 50 μm. *$P < 0.05$; **$P < 0.01$. Two-tailed Student's t test. Source data are provided in the Source Data File.

the lack of mechanical forces may be different from the ones underlying aging or hormone related osteoporosis. Our study demonstrates that PIEZO1 can coordinate the osteoblast–osteoclast crosstalk through directly sensing mechanical loading in osteoblast lineage cells. Our findings will inform therapeutic strategies for disuse osteoporosis in the setting of prolonged bed rest or exposure to long-term microgravity environment in space.

## Methods

**Ethics statement**. We have complied with all relevant ethical regulations for animal testing and research. All animal experiments were performed in the Animal Facility of Shanghai Institute of Biochemistry and Cell Biology and according to the protocol (approval number: SIBCB-NAF-14-001-S350-019) authorized by the Animal Care and Use Committee of Shanghai Institute of Biochemistry and Cell Biology, Chinese Academy of Sciences.

**Mouse lines**. *Piezo1^fl/fl^* mice (UCDAVIS KOMP repository, strain ID: Piezo1 tm1a) were crossed with the *Prx1^Cre^* strain (a gift from Andrew McMahon, Harvard University)[21], *Ctsk^Cre^* strain (provided by S. Kato, University of Tokyo)[46], *Dmp1^Cre^* strain (a gift from Jerry Q Feng, Baylor College of Medicine)[47] and *Col1^Cre ert2^* strain (a gift from Bin Zhou, Shanghai Institutes for Biological Sciences, Chinese Academy of Sciences)[48] to generate *Prx1^Cre^, Piezo1^fl/fl^* mice, *Ctsk^Cre^, Piezo1^fl/fl^* mice, *Dmp1^Cre^, Piezo1^fl/fl^* mice and *Col1^Cre ert2^, Piezo1^fl/fl^* mice. All mice analyzed were maintained on the C57BL/6 background. Animals were bred and maintained under specific pathogen free (SPF) conditions in the institutional animal facility of the Shanghai Institute of Biochemistry and Cell Biology, Chinese Academy of Sciences.

**Tail suspension model**. Tail suspension experiments to remove the body weight induced mechanical loading from hind limbs, mimicked the bone loss due to microgravity or disuse[24,25]. Six-week-old male mice were randomly divided into two groups for tail suspension and ground control. Mice were suspended in individual plastic cages for 7 days at about 30 degrees head-down tilt. The angle of suspension was adjusted to make sure that when the animal was fully stretching, their hind limbs were unable to touch the ground.

**Histological staining**. Tissues were fixed in 4% paraformaldehyde for 48 h and incubated in 15% DEPC-EDTA (PH7.8) for decalcification[49]. Then specimens were embedded in paraffin and sectioned at 6 μm. Tissue sections were used for TRAP staining according to the manufacturer's instructions (Sigma, 387A-1KT). Images were captured using a microscope (Olympus BX51, Tokyo, Japan).

**Immunofluorescence**. Freshly dissected bones were fixed in 4% paraformaldehyde for 48 h and incubated in 15% DEPC-EDTA (PH7.8) for decalcification. Then specimens were embedded in paraffin and sectioned at 6 μm. Sections were blocked in PBS with 10% horse serum for 1 h and then stained overnight with rabbit-anti-Collagen IX alpha 2, rabbit-anti-Collagen II alpha 1 or rabbit-anti-YAP. Donkey-anti-rabbit Cy3 was used as secondary antibody. DAPI (sigma, D8417) was used for counterstaining[50]. For cellular immunofluorescence studies, the indicated cells were seeded on glass slides and then fixed with 4% paraformaldehyde for 20 min. Subsequent steps were identical to those used for tissue immunofluorescence. Slides were mounted with anti-fade fluorescence mounting medium (Dako, S3023) and images were acquired with microscope (Olympus BX51, Tokyo, Japan) or confocal microscope (SP8, Leica).

**Calcein-Alizarin red S labeling**. Mice were injected intraperitoneally with 20 mg/ kg Calcein (1 mg/ml in 2% NaHCO₃ solution) and 40 mg/kg Alizarin red S (2 mg/ ml in H₂O) on day 0 and day 4 separately. On day 7, the mice were sacrificed and the bones were fixed and dehydrated and embedded with Embed-812 (Electron Microscopy Sciences). The samples were sectioned as 5 μm with hard tissue cutter. Consecutive sections were stained with silver nitrate to measure the bone morphology. The histomorphometric analysis was carried out semi-automatically with the OsteoMeasure image analyzer (OsteoMetric) according to recommended guidelines[51].

**Micro-quantitative computed tomography analysis**. The mouse femurs were skinned and fixed in 70% ethanol. Scanning was performed with the instrument micro CT system SkyScan1276 (Bruker, Kartuizersweg, Belgium) at a 9 μm resolution for quantitative analysis. The region from −40 to −240 slides below the growth plate was analyzed for trabecular number (Tb.N), trabecular thickness (Tb. Th), trabecular spacing (Tb.Sp), bone volume per tissue volume (BV/TV) and bone mineral density (BMD) with CTAn. The region from −400 to −456 slides below

the growth plate was analyzed for cortical thickness (Ct.Th) with CTAn. Three-dimensional reconstructions were created by stacking the two-dimensional images from the indicated regions with CTVox[52].

**RNA-seq and analysis**. Raw reads were mapped to mm9 using the TopHat version 1.4.1 program. We assigned FPKM (fragment per kilo base per million) as an expression value for each gene using Cufflinks version 1.3.0 software. The Cuffdiff software was used to identify differentially expressed genes between treatment and control samples. Differentially expressed gene heat maps were clustered by k-means clustering using the Euclidean distance as the distance and visualized using the Java TreeView software. The data were deposited into the GEO repository (GSE135282).

**Skeletal preparation and staining**. Mice were sacrificed by $CO_2$ and then removed skin and muscle. The remaining samples were transferred into acetone for 48 h after overnight fixation in 95% ethanol. Skeletons were then stained in 0.3% Alcian blue (Sigma, A5268) and 0.1% Alizarin red S (Sigma, A5533) solution (70% ethanol, 5% acetic acid)[52]. Specimens were incubated in 1% KOH until tissue was completely cleared and then kept in glycerol for pictures by Canon Camera.

**Calcium imaging**. Cells, isolated from WT and *Piezo1*-deficient mice, were seeded into 24-well cell slides (NEST), and incubated at 37 °C and 5% CO2. For $Ca^{2+}$ imaging, these cells were loaded with Fluo-4-AM ester (2 μM; Thermo Fisher Scientific) for 30 min, washed by HBSS (ThermoFisher, 14025076) for 3 times, and then rest for 15 min to allow dye de-esterification[53]. Then, the cells were exposed to Yoda1 (10 μM) using a flow chamber. The fluorescence intensity of Fluo-4-AM was monitored by live-cell imaging (UltraView VOX, PerkinElmer). Time-lapse fluorescence intensity measurements were analyzed using Image J.

**Flow cytometry**. Bone marrow cells were extracted from femurs and tibias of three WT and six CKO mice at 6-week. Marrow plugs were flushed with a 1-ml syringe using ice-cold DPBS (Corning, 21-031-CMR) containing 2% fetal bovine serum (FBS). These plugs were then dispelled into single cell and the red blood cells were removed by RBC lysis buffer for 5 min (Beyotime, C3702). Wash the cells twice by centrifugation at 600*g* for 5 min with ice-cold DPBS, 2% FBS. The cells were stained with eFluor 450 anti-CD31 (eBioscience,48-0311-80), PerCP/Cy5.5 anti-CD45 (BioLegend, 103132), APC/Cy7 anti-mouse TER-119 (BioLegend, 116223), FITC anti-mouse 6C3/Ly-51 (BioLegend, 108305), Brilliant Violet 605 anti-mouse CD90.2 (BioLegend, 140317), PE/Cy7 anti-mouse CD105 (BioLegend, 120409), APC anti-mouse CD200 (BioLegend, 123809) for 30 min. Then cells were measured by BC Cytoflex LX after twice of wash by centrifugation at 600*g* for 5 min with ice-cold DPBS, 2% FBS. The data were then analyzed with the CytExpert software.

**Cell differentiation**. BMSCs were extracted from femurs and tibias of 4- to 6-week-old mice. Marrow plugs were flushed with a 1-ml syringe using ice-cold DPBS (Corning, 21-031-CMR). These plugs were then dispelled into single cell and seeded in 10-cm dish with α-MEM (Corning, 10-022-CVR) (supplemented with 10% FBS, 1% Penicillin/Streptomycin). The non-adherent cells were removed and the adherent cells were used for BMSCs-derived osteoblastic cells.

For osteoblast differentiation, cells were cultured in α-MEM containing 10% FBS, 1% penicillin/streptomycin, 50 μg/ml ʟ-ascorbic acid (Sigma, A5960), and 1.08 mg/mL β-Glycerophosphate disodium salt hydrate (Sigma, G9422). Change the medium every 3 days. For quantitative analysis of ALP activity after 7 days of differentiation, osteoblasts were incubated with Alamar Blue (Invitrogen) for 1 h and read with a luminometer (Envision) for cell number quantification. Then remove the medium of osteoblasts and incubate with phosphatase substrate (Sigma, S0942) dissolved in the solution (6.5 mM $Na_2CO_3$, 18.5 mM $NaHCO_3$, 2 mM $MgCl_2$) for 20 min. Read by a luminometer (Envision) at A405 for ALP activity. The final ALP activity was normalized as A405/Alamar Blue. Bone nodule formation was stained with 1 mg/mL Alizarin red S solution (pH 5.5) after 21 days of induction[54].

For osteoclast differentiation, mouse bone marrow monocytes isolated from the femurs and tibias were seeded with $5 \times 10^5$ cells per well in a 96-well plate and cultured in α-MEM (supplemented with 10% FBS, 1% Penicillin/Streptomycin, 10 ng/ml M-CSF (PeproTech)) for 3 days. Subsequently, non-adherent cells were discarded and adherent cells were further cultured in the presence of 20 ng/ml M-CSF and 250 ng/ml soluble RANKL with indicated treatment. Cells were then fixed and stained for TRAP using a kit according to the manufacturer's instructions (Sigma, 387A-1KT)[54].

For osteoblasts and osteoclasts co-culture system, BMSCs-derived osteoblastic cells were seeded into a 24-well plate ($1 \times 10^5$ cells per well) or 96-well plate with bone biomimetic synthetic surface (Osteo Assay Surface, Corning, 3988) ($1 \times 10^4$ cells per well) with α-MEM and 10% FBS. Bone marrow monocytes were isolated from bone marrow and cultured for 24 h. The non-adherent bone marrow cells ($1 \times 10^7$ cells per well in a 24-well plate or $1 \times 10^6$ cells per well in a 96-well plate) were collected and cultured with BMSCs-derived osteoblastic cells in the presence of 1, 25-dihydroxyvitamin D3 (10 nM; Sigma) and PGE2 (1 μM; Sigma).The culture medium was half-changed every 2 days for 3 times. Cells were then fixed

and stained for TRAP using a kit according to the manufacturer's instructions (Sigma, 387A-1KT)[54]. For the pit resorption assay, the 96-well plate with bone biomimetic synthetic surface was treated with 10% NaClO solution after the co-culture. Resorption area was quantification using Image J.

**Collagen protein enrichment**. $3 \times 10^6$ 293T cells were seeded in 10-cm dish with DMEM (supplemented with 10% FBS and 1% penicillin-streptomycin) for 16 h and then transfected with 8 μg pyr1.1-flag-*Col2α1* or *Col9α2* or control plasmid. The supernatant was collected and incubated with 7 μL ANTI-FLAG M2 Affinity gel (Sigma, A2220), to enrich FLAG-COL2α1 or COL9α2 at 4°C overnight. And then the gel-binding FLAG-COL2α1 or COL9α2 or control was eluted by 40 μL FLAG Peptide (Sigma, F3290) diluted in TBS buffer (100 mM Tris, pH 7.5; 150 mM NaCl) after 5 times of wash by EBC buffer (50 mM Tris, pH 7.5, 120 mM NaCl and 0.5% NP-40) and once of wash by TBS buffer. The eluted collagen or control solution was then tested by Western Blot or used in the experiments of cell treatment in 1/100 dilution (COL2α1, 10 ng/mL; COL9α2, 20 ng/mL).

**Inhibitors treatment**. $5 \times 10^5$ C3H10T1/2 cells (ATCC, CRL-3268) were seeded in 6-well plates in α-MEM (supplemented with 10% FBS and 1% penicillin-strepto-mycin) for 48 h, and then change with the medium supplied with 10 μM XMU-MP-1 (Selleck, S8334) for 24 h. The cells were then for RNA extraction. For osteoclast differentiation or co-culture system, XMU-MP-1 was treated with 0.2 μM, SB273005 (Selleck, S7540) was treated with 50 nM and GRD peptides (Selleck, S8008) was treated with 20 μM.

**Mechanical compression devices**. For mechanical compression system, BMSCs-derived osteoblastic cells were seeded into hydrogel ($2 \times 10^6$ cells per well) with α-MEM (supplemented with 10% FBS and 1% penicillin-streptomycin) on BioFlex Compression Plates (Flexcell) for 24 h. Then, mechanical compression of these cells was generated in a humidified atmosphere of 5% $CO_2$ at 37 °C using Flexcell FX-5000 Compression System. The unit was used according to the manufacturer's protocol. A definite condition of 1 % intensity, 0.5 Hz, 4 h was applied to the cells to simulate moderate forces. In parallel as a control, cells were cultured on compression plates with the same conditions but without mechanical stress.

**Real-time RT-PCR analysis**. Total RNA was prepared using TRIzol (Sigma, T9424) and was reverse transcribed into cDNA with the PrimeScript RT Reagent Kit (Takara, PR037A). The real-time reverse transcriptase RT-PCR reaction was performed with the BioRad CFX96 system. The primer sets used were listed: *mBglap*-qF: GCAGCACAGGTCCTAAATAG; *mBglap*-qR:GGGCAATAAGGTAGTGAAC AG; *mPthlh*-qF: CATCAGCTACTGCATGA CAAGG; *mPthlh*-qR: GGTGGTTTTT GGTGTTGGGAG; *mMcsf*-qF: GGCTTGGCTTG GGATGATTCT; *mMcsf*-qR: GAGGGTCTGGCAGGTACTC; *mSema3a*-qF: GAAGAG CCCTTATGATCCCA AAC; *mSema3a*-qR: AGATAGCGAAGTCCCGTCCC; *mPiezo1*- qF: CTTACAC GGTTGCTGGTTGG; *mPiezo1*-qR: CACTTGATGAGGGCGGAAT; *mPiezo2*-qF: CATGACCTCTGCCTCCATCA; *mPiezo2*-qR: CCAGGGAAAGAAGCCG AACT; *mSost*-qF: GCCTCATCTGCCTACTTGTGC; *mSost*-qR: TCTGTCAGGAAGCG GGTGT; *mOpg*-qF: ACCCAGAAACTGGTCATCAGC; *mOpg*-qR:CTGCAATAC ACA CACTCATCACT; *mRankl*-qF: GCAGATTTGCAGGACTCGACT; *mRankl*-qR: CCCCA CAATGTGTTGCAGTT; *mCo9a1*-qF: CCCTGGTGCTCTTGGCT TAA; *mCol9a1*-qR:A CTTGTCCGGCCCTTTGC; *mCo9a2*-qF: GAGATGGGTCCT CGTGGCTAT; *mCol9a2*- qR: TCCTCTGGCACCTTGGCTC; *mCo2a1*-qF: CGGTC CTACGGTGTCAGG; *mCol2a1*-qR: GCAGAGGACATTCCCAGTGT; *mHprt*-qF: GTTAAGCAGTACAGCCCC AAA; *mHprt*-qR: AGGGCATATCCAACAACAA ACTT; *mRunx2*-qF: CCAACCGAGTCA TTTAAGGCT; *mRunx2*-qR: GCTCACGT CGCTCATCTTG; *mOsx*-qF: ATGGCGTCCT CTCTGCTTGA; *mOsx*-qR: GAAG GGTGGGTAGTCATTTG; *mAlp*-qF: CGGGACTG GTACTCGGATAA; *mAlp*-qR: ATTCCACGTCGGTTCTGTTC; *mCol1a1*-qF: GCTCCT CTTAGGGGCCACT; *mCol1a1*-qR: CCACGTCTTCACCATTGGGG; *mBsp*-qF: GACTT TTGAGTTAG CGGCACT; *mBsp*-qR: CCGCCAGCTCGTTTTCATC.

**Antibodies**. YAP antibody (4912) was purchased from CST. Rabbit polyclonal to PIEZO1 (ab128245) and rabbit polyclonal to Collagen II alpha 1 (ab34712) antibodies were purchased from Abcam. Collagen IX alpha 2 antibody (NBP2-30450) was purchased from Novus Biologicals.

**Statistics**. Statistical analysis was performed using the GraphPad Prism 8 software (GraphPad Software). Cell-based experiments were performed at least twice. Animals were randomized into different groups and at least 3 mice were used for each group, unless otherwise stated. The data are presented as mean ± SD. Two-tailed Student's *t* test was used to compare the effects of two groups, and ordinary one-way ANOVA was used to compare the effects of more than two groups. Statistically significant differences are indicated as follows: * for $p < 0.05$ and ** for $p < 0.01$.

**Reporting summary**. Further information on research design is available in the Nature Research Reporting Summary linked to this article.

## Data availability

Source data for Figs. 1–7 and Supplementary Figure 1, 2, 5, 6, 7, 9, 10 have been provided in the Source Data File. The RNA sequencing data have been deposited in the Gene Expression Omnibus (GEO) under accession GSE135282. All data and genetic material used for this paper are available from the authors on request.

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

## Acknowledgements

We thank Andrew McMahon for *Prx1Cre* mice, S. Kato for *CtskCre* mice, Jerry Q Feng for *Dmp1Cre* mice and Bin Zhou for *Col1Cre ert2* mice. We thank Marc N. Wein, Matthew B. Greenblatt and Jun Sun for their critical reading and helpful suggestions. We also thank the cell biology core facility, the animal core facility and the molecular biology core

facility of Shanghai Institute of Biochemistry and Cell Biology for assistance. This work was supported in part by grants from, the National Natural Science Foundation of China (NSFC) [81725010, 81672119, 81991512], the Strategic Priority Research Program of the Chinese Academy of Sciences [Grant No. XDB19000000] and the Space Medical Experiment Project of China Manned Space Program [HYZHXM01025]. W.Z. is a scholar of 'the National Science Fund for Distinguished Young Scholars' (NSFC) [81725010].

## Author contributions

L.W. and W.Z. conceived the study and wrote the manuscript. L.W. and X.Y. designed and performed experiments about the phenotype analysis of different mice and analyzed data. S.L. performed and analyzed the histomorphometry of $Prx1^{Cre}$; $Piezo1^{fl/fl}$ mice. L.Z. performed the compression system. N.W. and W.Z. contributed ideas and reviewed the manuscript.

## Competing interests

The authors declare no competing interests.
