## [Peer Review File · Nature Communications]

Reviewers' Comments:

Reviewer #1:

Remarks to the Author:

The current manuscript describes the consequences of PIEZO1 deletion on bone, using several conditional knockout animal models. The authors conclude that PIEZO1 is a skeletal mechanosensor in osteoblastic cells that regulate osteoclast activity via a pathway that involves YAP-collagen regulation. This is an interesting possibility. However, the study is not conclusive, and no mechanistic connections are shown between the different molecules. Further, the mechanism by which changes in collagen levels lead to increase osteoclast differentiation is not described or, at least, proposed.

Additional issues:

- 1- Abstract should be re-written to avoid mistakes, and to introduce YAP and the relevance of this molecule for the study.
- 2- The study by Grimston and colleagues (reference #9) does not describe the role of Cx43 hemichannels on bone mechanotransduction and therefore, the text on page 3 should be modified accordingly. Further 3 additional in vivo studies showed that Cx43 deletion actually enhances the response to mechanical loading. Therefore, the relevance of discussing the gap junction protein for the current study is not clear.
- 3- BMD stands for bone mineral density, not bone mass density.
- 4- Bones from the KO mice appear smaller, at least in the picture included in figure 1. How is the bone area of the mice compared to the control animals?
- 5- The differences between mice on calvaria bone should be detailed. The bones do not look particularly different in the images for supplementary figure 3a.
- 6- MAR, and not bone formation is shown. Therefore, the authors cannot make a conclusion about changes on bone formation rate between genotypes. Indeed, BFR/BS and MS/MS should also be included in the manuscript. In addition, the authors should indicate whether the measurements were done on the endocortical or periosteal surface.
- 7- Osteoclast number and surface, but not activity were measured throughout the study. Therefore, the authors cannot make any conclusion regarding osteoclast activity. Markers of bone resorption should be measured in vivo, and pit resorption activity should be measured in vivo to determine whether indeed osteoclasts are more active, or whether bone resorption is increased because there are more osteoclasts with similar activity.
- 8- Page 9, line 188: the nature of the mice described should be indicated here in the results section.
- 9- While the authors showed that both deletion of PIEZO1 and YAP result in decreased collagen levels, they do not provide any evidence that TAP is downstream of PIEZO1 in osteoblastic cells. So, even the relationship is suggested by the data, it is not demonstrated. The text should be modified accordingly.
- 10- References for the animal models should be provided in the methods section.
- 11- Statistical analysis is not appropriate when more than 2 groups are compared. Analysis should be re-made accordingly.
- 12- Controls should be included for the data described in figure 5 g,h. Thus, the effect of Piezo1 deletion in samples infected with control lentivirus should be tested in parallel with the effect of infecting collagen lentivirus.
- 13- Authors should discuss the potential mechanism by which reduced expression of certain collagens leads to increased osteoclast differentiation. Is there any previous evidence in this regard?

Reviewer #2:

Remarks to the Author:

This paper focuses on the role of mechanosensory Piezo channels in bone homeostasis. The authors report that Piezo1 deficiency in osteoblast leads to decreased bone mass and increased

fractures. The authors elegantly touch upon the subject of bone loss in microgravity environment or in an event of being bedridden. This aspect of the study will influence further development and thinking in the mechanotransduction field, bone formation and bone loss and the effective development of therapeutics. For the most part, the work is convincing. However, it was difficult to read due to long sentences and grammatical mistakes.

Please see below my comments to further clarify and strengthen the conclusions:

1. Supplementary figure 1C. There seems to be some discrepancy in Piezo expression in the skin. According to the literature Piezo2 expression in skin - Merkel cells is higher than Piezo1. In supplementary Figure 1C there is no Piezo2 expression in the skin. What type of skin (hairy or glabrous) or what type of skin cells are being used here?

2. Supplementary figure 1h. How are mice stimulated by Yoda1? Was there an IP or IV injections? It is not clear from the methods. Calcium imaging methods from cells and application of Yoda1 is clear but not clear how mice were treated with Yoda1, as mentioned in the figure legends.

3. Figure1: Was there any behavioral assay performed in the cage before sacrificing mice at 6 weeks? Were the knock out and control subjects were living normally in the cages? Was there any wheel or physical activity inside the cage? If there is an additional physical activity such as wheel or running on the treadmill, would the effect be similar or changed? It is worth mentioning whether the effect/phenotype could be rescued or negatively impacted with increased physical activity.

4. Figure 5 b and 6. It is worth checking LRP receptors, since they have a role in regulating bone mass.

5. There is no consistency throughout the study regarding sex of the mice. In Figure 1 and 4 the experiments are performed on males. In figure 3 experiments are performed on females. In figure 2 it is not mentioned whether they are all females or mixed population. It is best to be consistent. In my opinion compare females vs males, since human females are more affected by osteoporosis and hormone related bone disorders.

6. Typos and Grammatical Mistakes:

- Line 152: Firstly, we analyze (not Analysis)
- Line 155: "We further speculated that if PIEZO1 will impair osteoblast differentiation". Correct the grammar.
- Line 178: "indicating that indicating that".. repetition.
- Line 234-236. This long sentence needs grammatical correction. In its current form the message of this sentence is not clear.

Please find below a point-by-point response to the reviewers' comments.

Response to Reviewer1:

Reviewers' comments:

The current manuscript describes the consequences of PIEZO1 deletion on bone, using several conditional knockout animal models. The authors conclude that PIEZO1 is a skeletal mechanosensor in osteoblastic cells that regulate osteoclast activity via a pathway that involves YAP-collagen regulation. This is an interesting possibility. However, the study is not conclusive, and no mechanistic connections are shown between the different molecules. Further, the mechanism by which changes in collagen levels lead to increase osteoclast differentiation is not described or, at least, proposed.

We thank the reviewer for pointing out 'this is an interesting possibility'. We also appreciate the reviewer's thorough analysis about the mechanistic studies. In revision, we have performed additional experiments that yield new insights into the molecular mechanism through which PIEZO1 controls bone resorption.

For the mechanistic connections between PIEZO1 and YAP, in Fig. 6a-b of the revised MS, we showed that *Piezo1* deficiency led to decreased YAP nuclear localization. To further demonstrate the function of PIEZO1 depends on the regulation of YAP, we performed rescue experiments in osteoblast-osteoclast co-culture assays using XMU-MP-1, a YAP activator. As shown in the revised Fig. 6h-i, XMU-MP-1 inhibited the increased osteoclast differentiation caused by *Piezo1* deficiency, indicating that decreased YAP activity in osteoblastic cells is responsible for the phenotype of *Piezo1* deficiency. To exclude the function of XMU-MP-1 on osteoclasts, we also treated bone marrow monocytes with XMU-MP-1 during osteoclast differentiation, and found that XMU-MP-1 did not significantly affect osteoclast formation (revised Supplementary Fig. 9a-b).

For the mechanistic connections between PIEZO1 and collagen expression, in the revised MS, Fig.5f-h showed that *Piezo1* deficiency in osteoblastic cells led to decreased *Col2 α 1* and *Col9 α 2* expression. In addition, the expression levels of *Col2 α 1* (Fig. 5i) and *Col9 α 2* (Fig. 5j) were strongly increased by mechanical stimulation in WT cells, but not in *Piezo1*-deficient cells, demonstrating that PIEZO1 can sense the mechanical loading to regulate type II and IX collagen expression.

For the mechanistic connections between YAP and collagen expression, we showed that YAP directly regulates collagen expression (Fig. 6c-g).

Overall, we propose a model in which PIEZO1 in osteoblastic cells can sense mechanical loading and regulate the expression of different collagens through the regulation of YAP activity. However, how PIEZO1 directly regulates YAP activity is still open for the study of PIEZO1's activity. We discussed this issue in the discussion of this revised MS.

The reviewer also suggested that we propose a mechanism by which changes in

collagen levels in osteoblastic cells lead to increased osteoclast differentiation. We fully agree with the reviewer that this is an important question. Previous work has proposed that the intermediate metabolite of type II collagen could inhibit osteoclast differentiation and activity by an RGD-mediated mechanism². Disrupted type IX collagen has also been reported to cause abnormal nanotopography of bone and increased osteoclast activity³. These studies provide insights in understanding how non-type I collagens can regulate osteoclastogenesis. However, the receptors on the osteoclasts responding to collagens stimulation remain to be further elucidated. As previously reported, integrin can regulate osteoclast differentiation and function^{4, 5, 6}. We propose that collagens may function through integrin to repress osteoclastogenesis. To test this hypothesis, we enriched the COL2 α 1 and COL9 α 2 proteins from the supernatants of *Col2 α 1* or *Col9 α 2* overexpressing 293T cells and treated bone marrow monocytes with these enriched proteins. We found that both COL2 α 1 and COL9 α 2 proteins could inhibit osteoclastogenesis (revised Supplementary Fig. 10a-c). Furthermore, the repression of COL2 α 1 or COL9 α 2 proteins on osteoclast formation was inhibited by integrin inhibitors SB273005 or RGD peptides (revised Supplementary Fig. 10d-g), demonstrating that the inhibition of osteoclast differentiation by COL2 α 1 and COL9 α 2 proteins depends on integrin pathway. We discussed the above data in the discussion of the revised MS and propose that these collagens could regulate osteoclast activity through inhibiting integrin pathway.

Additional issues:

1- Abstract should be re-written to avoid mistakes, and to introduce YAP and the relevance of this molecule for the study.

We thank the reviewer for this kind suggestion. We have checked and corrected the mistakes in the abstract and introduced the relationship of YAP and mechanotransduction in the revised manuscript line 43 and 70.

2- The study by Grimston and colleagues (reference #9) does not describe the role of Cx43 hemichannels on bone mechanotransduction and therefore, the text on page 3 should be modified accordingly. Further 3 additional in vivo studies showed that Cx43 deletion actually enhances the response to mechanical loading. Therefore, the relevance of discussing the gap junction protein for the current study is not clear.

We thank the reviewer for pointing out this issue. We have added the additional references and re-written the function of Cx43 on mechanotransduction in the revised manuscript line 65.

3- BMD stands for bone mineral density, not bone mass density.

Thanks the reviewer for pointing out this issue. We have corrected it in the revised manuscript.

4- Bones from the KO mice appear smaller, at least in the picture included in figure 1. How is the bone area of the mice compared to the control animals?

As the reviewer pointed out, the long bones of *Prx1^{Cre}*; *Piezo1^{fl/fl}* mice were smaller

than WT mice (revised Supplementary Fig. 2g). In addition, we analyzed the bone surfaces of WT and *Prx1^{Cre}; Piezo1^{fl/fl}* mice by μ -CT. Both cortical and trabecular bone surfaces of *Prx1^{Cre}; Piezo1^{fl/fl}* mice were significantly decreased, compared with the WT mice (revised Supplementary Fig. 2h-i), further confirming decreased bone mass.

Supplementary Fig. 2 g X-ray images of the femurs from 6-week-old male WT and *Prx1^{Cre}; Piezo1^{fl/fl}* mice. Representative images for 6 independent samples. **h-i** μ -CT analysis of the distal femurs from 6-week-old male WT and *Prx1^{Cre}; Piezo1^{fl/fl}* mice for cortical bone surface (Ct.BS) (**h**) and trabecular bone surface (Tb.BS) (**i**). Two-tailed Student's t-test. Data are mean \pm s.d., n = 6.

5- The differences between mice on calvaria bone should be detailed. The bones do not look particularly different in the images for supplementary figure 3a.

We thank the reviewer for pointing out this issue. Calvarial bones from WT and *Prx1^{Cre}; Piezo1^{fl/fl}* mice were indistinguishable, as demonstrated by both μ -CT 3D image and alcian blue-alizarin red staining (revised Supplementary Fig.4a, b). In addition, the TRAP staining of skulls did not show significant differences between WT and *Prx1^{Cre}; Piezo1^{fl/fl}* mice (revised Supplementary Fig.7a).

Supplementary Fig. 4 **a** 3D μ -CT images of the cranial bones isolated from 3-week-old male WT and *Prx1^{Cre}, Piezo1^{fl/fl}* mice. Representative images for 3 independent samples. **b** Alcian blue and Alizarin red S staining of the skulls of WT and *Prx1^{Cre}, Piezo1^{fl/fl}* mice at postnatal day 3. Representative images for more than 3 independent samples.

Supplementary Fig. 7 **a** Whole-mount TRAP staining for the skulls of 3-week-old male mice. Representative images for 3 independent samples.

6- MAR, and not bone formation is shown. Therefore, the authors cannot make a conclusion about changes on bone formation rate between genotypes. Indeed, BFR/BS and MS/MS should also be included in the manuscript. In addition, the authors should indicate whether the measurements were done on the endocortical or periosteal surface.

We thank the reviewer for this kind suggestion. To answer this question, we re-analyzed the Alizarin red S and Calcein double-labeling samples of WT and *Prx1^{Cre}, Piezo1^{fl/fl}* mice for the BFR/BS and MS/BS of endo-cortical and trabecular bone. Both BFR/BS and MS/BS were not significantly changed in *Prx1^{Cre}, Piezo1^{fl/fl}* mice (revised Fig. 2j-k). To further confirm this conclusion, we analyzed the bone formation marker PINP by ELISA, and found that the PINP levels were not significantly changed in *Prx1^{Cre}, Piezo1^{fl/fl}* mice compared to WT controls (revised Fig. 2l).

Fig. 2 j-k Quantification of bone formation rate normalized to bone surface (BFR/BS) (**j**) and mineralizing surface normalized to bone surface (MS/BS) (**k**) of endo-cortical and trabecular bones of proximal tibias from 6-week-old male WT and *Prx1^{Cre}, Piezo1^{fl/fl}* mice. **P* < 0.05. Two-tailed Student's t-test. Data are mean ± s.d., n=6. **l** Serum PINP levels in 6-week-old male WT and *Prx1^{Cre}, Piezo1^{fl/fl}* mice. **P* < 0.05; ***P* < 0.01. Two-tailed Student's t-test. Data are mean ± s.d., WT, n=6, CKO, n=8.

7- Osteoclast number and surface, but not activity were measured throughout the study. Therefore, the authors cannot make any conclusion regarding osteoclast activity. Markers of bone resorption should be measured *in vivo*, and pit resorption activity should be measured *in vivo* to determine whether indeed osteoclasts are more active, or whether bone resorption is increased because there are more osteoclasts with similar activity.

We thank the reviewer for this kind suggestion. In our study, we demonstrated that the osteoclast number and surface were increased in *Prx1^{Cre}; Piezo1^{fl/fl}* mice (revised Fig. 2n and 2o). To further determine the bone resorption activity of osteoclasts *in vivo*, we analyzed the eroded surface (ES/BS) by histomorphometry and the serum bone resorption marker CTX-I by ELISA (revised Fig. 2m and 2p). Both eroded surface (ES/BS) and CTX-I were significantly increased in *Prx1^{Cre}; Piezo1^{fl/fl}* mice (revised Fig. 2m and 2p), demonstrating increased bone resorption *activity* of osteoclasts. As the reviewer suggested, we performed pit resorption assays by co-culturing bone marrow monocytes with the BMSCs-derived osteoblasts from WT and *Piezo1*-deficient mice. *Piezo1*-deficient mice showed higher pit resorption activity, determined by resorption area (revised Fig. 5c-d). These data indicate that osteoclast activity is increased in *Piezo1*-deficient mice. The data is consistent with the previous report that Collagen 9 can inhibit osteoclast activity. We added this part of data into the main figures in the revised MS.

Fig. 2 m-n Histomorphometric analysis of the images from (c) for eroded surface per bone surface (ES/BS) (m), number of osteoclasts per bone perimeter (N.Oc/B.Pm) (n) and osteoclast surface per bone surface (Oc.S/BS) (o). *P < 0.05; **P < 0.01. Two-tailed Student's t-test. Data are mean \pm s.d., WT, n=4, CKO, n=3. **p** Serum CTX-I levels in 6-week-old male WT and *Prx1^{Cre}, Piezo1^{fl/fl}* mice. *P < 0.05; **P < 0.01. Two-tailed Student's t-test. Data are mean \pm s.d., WT, n=6, CKO, n=8.

Fig. 5 c TRAP staining of the osteoclasts (top panel) and resorption area (bottom panel) in the co-culture system on the bone biomimetic synthetic surface. **d** Quantification of the resorption area. *P < 0.05; **P < 0.01. Two-tailed Student's t-test. Data are mean \pm s.d., n=6.

8- Page 9, line 188: the nature of the mice described should be indicated here in the results section.

We thank the reviewer for this kind suggestion. We have described that the mice were *Prx1^{Cre}; Piezo1^{fl/fl}* mice in the revised manuscript line 200.

9- While the authors showed that both deletion of PIEZO1 and YAP result in decreased collagen levels, they do not provide any evidence that YAP is downstream of PIEZO1 in osteoblastic cells. So, even the relationship is suggested by the data, it is not demonstrated. The text should be modified accordingly.

We thank the reviewer for this kind suggestion. To demonstrate YAP is downstream of PIEZO1 in osteoblastic cells, we showed that *Piezo1* deficiency led decreased YAP

nuclear localization (revised Fig. 6a-b). We also performed rescue experiments in the osteoblast-osteoclast co-culture assay with XMU-MP-1, a YAP activator. We found that XMU-MP-1 could inhibit the increased osteoclast differentiation caused by *Piezo1* deficiency in the co-culture system (revised Fig. 6h-i). To avoid the function of XMU-MP-1 on osteoclasts, we also treated bone marrow monocytes with XMU-MP-1 during osteoclast differentiation, and found that XMU-MP-1 did not significantly affect the osteoclasts (revised Fig. S9a, b). These data indicate that decreased YAP activity in osteoblastic cells, but not in bone marrow monocytes is responsible for the phenotype of *Piezo1* deficiency.

Fig. 6 h-i Osteoclastogenesis by OB-OC co-culture in vitro using BMSCs-derived osteoblasts from WT and *Prx1^{Cre}, Piezo1^{fl/fl}* mice, the co-culture system was treated with DMSO or XMU-MP-1 (0.2μM), respectively. **h** TRAP staining of the osteoclasts in the co-culture system. Scale bar = 100μm. **i** Co-culture supernatants were measured for TRAP activity via colorimetric readout (A405). *P < 0.05; **P < 0.01. Ordinary one-way ANOVA. Data are mean ± s.d., n=4.

Supplementary Fig. 9 a Bone marrow monocytes were seeded in 96-well plates and treated with 20 ng/ml M-CSF and 250 ng/ml RANKL for 6 days with DMSO or XMU-MP-1 (0.2μM). **b** Quantification of the osteoclast differentiation by TRAP activity of the culture supernatants. *P < 0.05; **P < 0.01. Two-tailed Student's t-test. Data are mean ± s.d., n=6.

10- References for the animal models should be provided in the methods section.

We thank the reviewer for this kind suggestion. And we have provided references for the animal models in the method section in the revised manuscript line 369-388.

11- Statistical analysis is not appropriate when more than 2 groups are compared. Analysis should be re-made accordingly.

Per the Reviewer's kind suggestions, we re-made statistical analysis using ordinary one-way ANOVA when more than two groups were compared.

12- Controls should be included for the data described in figure 5 g,h. Thus, the effect of *Piezo1* deletion in samples infected with control lentivirus should be tested in parallel with the effect of infecting collagen lentivirus.

We thank the reviewer for this kind suggestion. We have included the controls and described the effect of GFP control lentivirus on *Piezo1*-deficient cells in the revised manuscript line 252-257.

13- Authors should discuss the potential mechanism by which reduced expression of certain collagens leads to increased osteoclast differentiation. Is there any previous evidence in this regard?

We thank the reviewer for this kind suggestion. We fully agree with the reviewer that this is an important question. Previous study has proposed that the intermediate metabolite of type II collagen could inhibit osteoclast differentiation and activity by an RGD-mediated mechanism². Disrupted type IX collagen has been also reported to cause abnormal nanotopography of bone and increased osteoclast activity³. These studies may provide insights in understanding how the collagens regulate osteoclastogenesis. We cited these publications in the revised manuscript line 250 and line 311-321. However, the mechanism or receptors of the osteoclasts responding to collagens stimulation remain to be further elucidated. As previously reported, integrin can regulate osteoclasts differentiation and function^{4, 5, 6}. We propose that integrin could mediate the repression of collagens on osteoclastogenesis. To test this hypothesis, we enriched the COL2 α 1 and COL9 α 2 proteins from the supernatants of *Col2 α 1* and *Col9 α 2* overexpressing 293T cells and treated the bone marrow monocytes during osteoclast differentiation and found that both COL2 α 1 and COL9 α 2 proteins could inhibit osteoclastogenesis (revised Supplementary Fig. 10a-c), consistent with the co-culture experiment (revised Fig. 5k-l). However, COL2 α 1 and COL9 α 2 proteins could not repress the osteoclast formation when the cells were treated with integrin inhibitors SB273005 or RGD peptides (revised Supplementary Fig. 10d-g), indicating that these two collagens' inhibition on osteoclast differentiation depends on integrin pathway.

Supplementary Fig. 10 **a** Analysis of COL2a1 and COL9a2 enriched from the supernatants of 293T cells transfected with control, *Col2a1* or *Col9a2* plasmids by coomassie blue staining (left) or western blot of FLAG antibody (right). **b** Bone marrow monocytes were seeded in 96-well plates and treated with 20 ng/ml M-CSF and 250 ng/ml RANKL for 6 days with indicated collagens or control. Scale bar = 100 μ m. **c** Quantification of the osteoclast differentiation by TRAP activity of the culture supernatants from (**b**). *P < 0.05; **P < 0.01. Ordinary one-way ANOVA. Data are mean \pm s.d., n=10. **d-e** Bone marrow monocytes were seeded in 96-well plates and treated with 20 ng/ml M-CSF and 250 ng/ml RANKL for 6 days with indicated treatment. Scale bar = 100 μ m. **f-g** Quantification of the osteoclast differentiation by TRAP activity of the culture supernatants from (**d-e**). *P < 0.05; **P < 0.01. Ordinary one-way ANOVA. Data are mean \pm s.d., n=4.

Reviewer #2 (Remarks to the Author):

This paper focuses on the role of mechanosensory Piezo channels in bone homeostasis. The authors report that Piezo1 deficiency in osteoblast leads to decreased bone mass and increased fractures. The authors elegantly touch upon the subject of bone loss in microgravity environment or in an event of being bedridden. This aspect of the study will influence further development and thinking in the mechanotransduction field, bone formation and bone loss and the effective development of therapeutics. For the most part, the work is convincing. However, it was difficult to read due to long sentences and grammatical mistakes.

We thank the reviewer for carefully considering our manuscript and the positive comments. We also thank the reviewer for pointing out some issues relate to the writing. We have shortened some sentences and corrected grammatical mistakes to make the paper easier to read.

Please see below my comments to further clarify and strengthen the conclusions:

1. Supplementary figure 1C. There seems to be some discrepancy in Piezo expression in the skin. According to the literature Piezo2 expression in skin - Merkel cells is higher than Piezo1. In supplementary Figure 1C there is no Piezo2 expression in the skin. What type of skin (hairy or glabrous) or what type of skin cells are being used here?

We thank the reviewer for pointing out this question. In our study, hairy skin tissue was used to define the expression profiles of *Piezo1* and *Piezo2*. As the reviewer suggested, *Piezo2* plays important roles in Merkel cells in the skin. In our data, *Piezo2* is expressed in skin, but lesser than in DRG (Fig. 1c). In addition, *Piezo2* is highly expressed in DRG, and less in bone cells and skin, as reported previously (Fig. R1a-b)¹.

Figure R1. **a-b** mRNA expression profiles of (a) *Piezo1* and (b) *Piezo2* determined by means of quantitative PCR from various adult mouse tissues.¹

2. Supplementary figure 1h. How are mice stimulated by Yoda1? Was there an IP or IV injections? It is not clear from the methods. Calcium imaging methods from cells and application of Yoda1 is clear but not clear how mice were treated with Yoda1, as mentioned in the figure legends.

We thank the reviewer for pointing out this indistinct description. We cultured BMSCs-derived osteoblasts which were isolated from WT and *Prx1^{Cre}*, *Piezo1^{fl/fl}* mice, then treated the cells with Yoda1 for calcium imaging. We made revisions in the figure legend and in the methods to make the description clear.

3. Figure1: Was there any behavioral assay performed in the cage before sacrificing mice at 6 weeks? Were the knock out and control subjects were living normally in the cages? Was there any wheel or physical activity inside the cage? If there is an additional physical activity such as wheel or running on the treadmill, would the effect be similar or changed? It is worth mentioning whether the effect/phenotype could be rescued or negatively impacted with increased physical activity.

We thank the reviewer for this suggestion. In our experiments, mice were living normally in cages, and there was no wheel or physical activity inside the cages. To determine the function of PIEZO1 as a mechanical loading sensor, we performed tail suspension experiments to remove mechanical loading from hind limbs (Fig. 4a-e),

mimicking the bone loss due to microgravity or disuse^{7, 8, 9}. As expected, tail suspension induced bone loss in the distal femurs of WT but not PIEZO mutant mice (Fig. 4a-f). The reviewer suggested to check whether the effect/phenotype could be rescued or negatively impacted with increased physical activity. We fully agree with the reviewer that the data from additional physical activity on the treadmill or wheel is interesting and would further solidify our hypothesis that PIEZO1 can function as a mechanical sensor for increased physical activity. Actually, we found that the expression levels of *Piezo1* were increased after running on the treadmill (Fig. R2a-c), indicating that PIEZO1 could respond to the increased physical activity. More interestingly, we found that increased physical activity can promote the expression of *Col2α1* and *Col9α2*, indicating that the correlation between the PIEZO1 expression with collagens expression. However, considering the time constraints for resubmission of this manuscript and the competitive nature of this field, we prefer to report these results in our future following publication. We are planning these future experiments by breeding more mice.

Figure R2. **a-c** Indicated gene expression analysis of the whole bone of control or exercise mice with treadmill for 6 weeks.

4. Figure 5 b and 6. It is worth checking LRP receptors, since they have a role in regulating bone mass.

We thank the reviewer for this suggestion. We checked the LRP receptors in the RNA-seq data of cortical bones of WT and *Prx1^{Cre}; Piezo1^{fl/fl}* mice, and found that the LRP receptors were not significantly changed (Fig. R3a), indicating that the regulation of bone mass by PIEZO1 is independent of the regulation of the expression of LRP receptors. As the reviewer suggested, we checked LRP receptors expression levels in YAP knock down samples (Fig. R3b). LRP9 and LRP11 expression were regulated by YAP knock down (Fig. R3b). However, the function of LRP9/11 on bone and the relationship with YAP have not been reported previously and remain to be further investigated.

Figure R3. **a** Indicated gene expression analysis of cortical bone of WT and *Prx1^{Cre}*, *Piezo1^{fl/fl}* mice. **b** Indicated gene expression analysis in C3H10 cells infected with shControl and shYAP lenti-virus.

5. There is no consistency throughout the study regarding sex of the mice. In Figure 1 and 4 the experiments are performed on males. In figure 3 experiments are performed on females. In figure 2 it is not mentioned whether they are all females or mixed population. It is best to be consistent. In my opinion compare females vs males, since human females are more affected by osteoporosis and hormone related bone disorders.

We thank the reviewer for pointing out this issue. We did quantitative computed tomography (μ -QCT) analysis in both male and female WT and *Prx1^{Cre}*; *Piezo1^{fl/fl}* mice. As expected, bone mass of female mice is less than male mice. However, bone mass was comparably decreased in both male and female *Prx1^{Cre}*; *Piezo1^{fl/fl}* mice compared to WT controls of the same gender (revised Supplementary Fig. 2a), with the decreased trabecular bone volume (BV/TV, revised Supplementary Fig. 2b), trabecular number (Tb. N, revised Supplementary Fig. 2c), cortical thickness (Ct. Th, revised Supplementary Fig. 2f) and increased trabecular spacing (Tb. Sp, revised Supplementary Fig. 2e). Trabecular thickness (Tb. Th, revised Supplementary Fig. 2d) was not changed significantly in female *Prx1^{Cre}*; *Piezo1^{fl/fl}* mice. Overall, we propose that PIEZO1 mediating mechanotransduction in osteoblastic cells is independent of gender. Following the kind suggestions from the reviewer, we have clarified the gender of mice in all experiments.

Supplementary Fig. 2 a 3D μ -CT images of trabecular bones of distal femurs isolated from 6-week-old female WT and *Prx1^{Cre}, Piezo1^{fl/fl}* mice. **b-f** μ -CT analysis of distal femurs from 6-week-old female WT and *Prx1^{Cre}, Piezo1^{fl/fl}* mice for bone volume per tissue volume (BV/TV) (**b**), trabecular number (Tb.N) (**c**), trabecular thickness (Tb. Th) (**d**), trabecular separation (Tb. Sp) (**e**) and cortical thickness (Ct.Th) of middle shaft of femurs (**f**). *P < 0.05; **P < 0.01. Two-tailed Student's t-test. Data are mean \pm s.d., n = 6.

6. Typos and Grammatical Mistakes:

- Line 152: Firstly, we analyze (not Analysis)
- Line 155: "We further speculated that if PIEZO1 will impair osteoblast differentiation". Correct the grammar.
- Line 178: "indicating that indicating that" repetition.
- Line 234-236. This long sentence needs grammatical correction. In its current form the message of this sentence is not clear.

We thank the reviewer for pointing out these typos and grammatical mistakes. We have corrected all of these mistakes in the revised manuscript.

Reference:

1. Coste B, Mathur J, Schmidt M, Earley TJ, Ranade S, Petrus MJ, *et al.* Piezo1 and Piezo2 are essential components of distinct mechanically activated cation channels. *Science* 2010, **330**(6000): 55-60.
2. Hayashi S, Wang Z, Bryan J, Kobayashi C, Faccio R, Sandell LJ. The type II collagen N-propeptide, PIIBNP, inhibits cell survival and bone resorption of osteoclasts via integrin-mediated signaling. *Bone* 2011, **49**(4): 644-652.
3. Wang CJ, Iida K, Egusa H, Hokugo A, Jewett A, Nishimura I. Trabecular bone

deterioration in col9a1+/- mice associated with enlarged osteoclasts adhered to collagen IX-deficient bone. *Journal of bone and mineral research : the official journal of the American Society for Bone and Mineral Research* 2008, **23**(6): 837-849.

4. Tucci M, De Palma R, Lombardi L, Rodolico G, Berrino L, Dammacco F, *et al.* beta(3) Integrin subunit mediates the bone-resorbing function exerted by cultured myeloma plasma cells. *Cancer research* 2009, **69**(16): 6738-6746.
5. Zou W, Teitelbaum SL. Absence of Dap12 and the alphavbeta3 integrin causes severe osteopetrosis. *The Journal of cell biology* 2015, **208**(1): 125-136.
6. Ross FP, Teitelbaum SL. alphavbeta3 and macrophage colony-stimulating factor: partners in osteoclast biology. *Immunological reviews* 2005, **208**: 88-105.
7. Chowdhury P, Long A, Harris G, Soulsby ME, Dobretsov M. Animal model of simulated microgravity: a comparative study of hindlimb unloading via tail versus pelvic suspension. *Physiological reports* 2013, **1**(1): e00012.
8. Sakai A, Nakamura T. Changes in trabecular bone turnover and bone marrow cell development in tail-suspended mice. *Journal of musculoskeletal & neuronal interactions* 2001, **1**(4): 387-392.

Reviewers' Comments:

Reviewer #1:

Remarks to the Author:

While the manuscript has improved substantially, the statement regarding Cx43 is still incorrect, since Cx43 hemichannels are not proteins, and there is no evidence that Cx43 has a beneficial role in mechanical loading in vivo

Reviewer #2:

Remarks to the Author:

The authors have provided satisfactory answers to all my concerns in a point by point rebuttal letter. No further comments from this reviewer.

A point-by-point response to the reviewers.

Thanks the reviewers for the suggestions for us to improve the manuscript.

Reviewer #1 (Remarks to the Author):

While the manuscript has improved substantially, the statement regarding Cx43 is still incorrect, since Cx43 hemichannels are not proteins, and there is no evidence that Cx43 has a beneficial role in mechanical loading in vivo.

Thanks the reviewer for raising this question. The opening of Connexin43 (Cx43) hemichannel depends on interaction with integrin in response to shear stress in osteocytes¹. However, Cx43 deficient mice showed an enhanced anabolic response to mechanical load². We have rewritten this part in the revised manuscript line 49-52.

Reviewer #2 (Remarks to the Author):

The authors have provided satisfactory answers to all my concerns in a point by point rebuttal letter. No further comments from this reviewer.

Thanks the reviewer for the suggestions for us to improve the manuscript.

1. Batra N, Burra S, Siller-Jackson AJ, Gu S, Xia X, Weber GF, *et al.* Mechanical stress-activated integrin alpha5beta1 induces opening of connexin 43 hemichannels. *Proc Natl Acad Sci U S A* 2012, **109**(9): 3359-3364.
2. Zhang Y, Paul EM, Sathyendra V, Davison A, Sharkey N, Bronson S, *et al.* Enhanced osteoclastic resorption and responsiveness to mechanical load in gap junction deficient bone. *PLoS one* 2011, **6**(8): e23516.